# ALKBH5 controls the meiosis-coupled mRNA clearance in oocytes by removing the $N^6$-methyladenosine methylation

Long Bai [1,2,3,4,5] ✉, Yu Xiang[1,2,3,4], Minyue Tang[1,2,3,4], Shuangying Liu[1,2,3], Qingqing Chen[1,2,3], Qichao Chen[1,2,3], Min Zhang [1,2,3], Shan Wan[1,2,3], Yimiao Sang[1,2,3], Qingfang Li[1,2,3], Sisi Wang[1,2,3], Zhekun Li[1,2,3], Yang Song[1,2,3], Xiaoling Hu[1,2,3], Luna Mao[1,2,3], Guofang Feng[1,2,3], Long Cui [1,2,3], Yinghui Ye[1,2,3] & Yimin Zhu [1,2,3,5] ✉

$N^6$-methyladenosine (m⁶A) maintains maternal RNA stability in oocytes. One regulator of m⁶A, ALKBH5, reverses m⁶A deposition and is essential in RNA metabolism. However, the specific role of ALKBH5 in oocyte maturation remains elusive. Here, we show that *Alkbh5* depletion causes a wide range of defects in oocyte meiosis and results in female infertility. Temporal profiling of the maternal transcriptomes revealed striking RNA accumulation in *Alkbh5⁻/⁻* oocytes during meiotic maturation. Analysis of m⁶A dynamics demonstrated that ALKBH5-mediated m⁶A demethylation ensures the timely degradation of maternal RNAs, which is severely disrupted following *Alkbh5⁻/⁻* depletion. A distinct subset of transcripts with persistent m⁶A peaks are recognized by the m⁶A reader IGF2BP2 and thus remain stabilized, resulting in impaired RNA clearance. Additionally, reducing IGF2BP2 in *Alkbh5*-depleted oocytes partially rescued these defects. Overall, this work identifies ALKBH5 as a key determinant of oocyte quality and unveil the facilitating role of ALKBH5-mediated m⁶A removal in maternal RNA decay.

The intricately orchestrated transition from oocyte to embryo is accompanied by dynamic changes in maternal transcripts[1]. In growing oocytes, maternal RNAs are robustly transcribed and stored to support further oocyte maturation and embryo development. Once oocytes develop to the full-grown germinal vesicle (GV) stage, extensive but target-selective maternal RNA decay is triggered by gonadotropin, and de novo transcription ceases. Transcription does not resume until zygotic genome activation (ZGA) after fertilization. Consequently, by the 2-cell stage, approximately 90% of cellular RNAs in mouse embryos are degraded[2]. Therefore, the maintenance and timely decay of maternal transcripts is a prerequisite for the acquisition of full developmental competence in oocytes. Indeed, oocytes that undergo delayed RNA decay exhibit a reduced polar body extrusion (PBE) rate, distorted spindles and embryo development arrest[3,4]. These observations raise intriguing questions regarding how mRNA decay is regulated during meiotic maturation.

Posttranscriptional modifications have been demonstrated to play essential roles in regulating maternal transcripts, including in stability maintenance of transcript stability, splicing and nuclear export[5,6], thus determining RNA fate. Notable among these modifications is $N^6$-methyladenosine (m⁶A), which is one of the most abundant modifications on eukaryotic RNA and is found in a wide range of

[1]Department of Reproductive Endocrinology, Women's Hospital, School of Medicine, Zhejiang University, Hangzhou, Zhejiang 310006, China. [2]Key Laboratory of Reproductive Genetics (Ministry of Education), Women's Hospital, Zhejiang University School of Medicine, Hangzhou, Zhejiang 310006, China. [3]Women's Reproductive Health Laboratory of Zhejiang Province, Women's Hospital, Zhejiang University School of Medicine, Hangzhou, Zhejiang 310006, China. [4]These authors contributed equally: Long Bai, Yu Xiang, Minyue Tang. [5]These authors jointly supervised this work: Long Bai, Yimin Zhu. ✉e-mail: bailong0375@zju.edu.cn; zhuyim@zju.edu.cn

species[7,8]. Its deposition on RNA is catalyzed by the METTL3-METTL14-WTAP methyltransferase complex and other regulatory factors, including KIAA1429 and ZC3H13 (known as "writers"), while it can be removed by two demethylases, FTO and ALKBH5 (known as "erasers")[7]. An expanding set of m⁶A-binding proteins (also known as "readers"), such as YTHDF2 and IGF2BP2, recognize m⁶A -modified RNAs to implement their functions in regulating cellular physiology[7,8]. This epigenetic mark affects almost every aspect of RNA metabolism and exerts profound effects on a broad range of biological processes ranging from organ development to cancer.

Previous studies have highlighted the indispensable functions of m⁶A readers and writers in oogenesis[9]. For example, oocyte-specific loss of the m⁶A reader *Ythdf2* contributes to incompetence in metaphase II (MII) oocytes leading to failure to support early zygotic development[10]. In addition, YTHDC1, another nuclear m⁶A reader, is crucial to germline development in mice, as *Ythdc1*-deficient oocytes are blocked at the primary follicle stage[11]. Similarly, defective follicle development is also detected in mice depleted of key m⁶A methyltransferases, namely, KIAA1429 and METTL3, resulting in female infertility[12–15]. Remarkably, recent advances in low-input m⁶A analysis technologies have revealed a unique m⁶A landscape in murine oocyte maturation[16–18]. Integrative analysis of the transcriptome and m⁶A methylome at single oocyte resolution suggest that m⁶A acts a protective factor to ensure the stability of maternal RNA in oocytes. These findings collectively suggest that m⁶A is deposited on maternal RNA and determines its fate, thereby affecting oocyte competence. However, while most studies have focused on the functions and consequences of RNA m⁶A methylation induced and mediated by writer and reader proteins, respectively, much less is known about the role of m⁶A removal, catalyzed by eraser proteins, in oocyte maturation.

As one of the known m⁶A demethylases, alpha-ketoglutarate-dependent dioxygenase alkB homolog 5 (ALKBH5) erases m⁶A modifications from RNA and has been shown to have roles in mammalian development and human diseases[19]. Male mice with *Alkbh5* deficiency are subfertile and display impaired spermatogenesis[20]. Further analysis of the transcriptome and m⁶A profiles in spermatogenic cells has revealed that appropriate m⁶A demethylation by ALKBH5 is required for controlling the splicing and stability of mRNA in spermatogenesis[21]. Aside from its roles in male fertility, ALKBH5 appears to be involved in the regulation of oocyte development, as female *Alkbh5⁻/⁻* mice also show impaired fertility[20]. However, it remains unknown how ALKBH5 affects female fertility and whether the demethylation function of ALKBH5 is involved.

To address these questions, we generated an *Alkbh5* total knockout (*Alkbh5⁻/⁻*) mouse strain and investigated the potential role of ALKBH5 in female fertility and oocyte developmental competence. Our results demonstrated that *Alkbh5⁻/⁻* female mice are infertile and that the oocyte maturation process in these mice is compromised. Mechanistically, we found that ALKBH5-mediated m⁶A removal ensures timely RNA decay during oocyte meiotic maturation. Conversely, the altered m⁶A modifications caused by *Alkbh5* loss are recognized by m⁶A reader protein IGF2BP2 gene-specific.

## Results

### ALKBH5 depletion results in developmental arrest of oocytes

ALKBH5 is well known to be colocalized with nuclear speckles in HeLa cells[20], thus influencing mRNA processing. Consistently, we observed a diffuse nucleoplasmic pattern of ALKBH5 expression in oocytes during folliculogenesis (Supplementary Fig. 1a). To further detect its subcellular localization, in vitro transcribed *Alkbh5*-Flag-eGFP mRNA was injected into oocytes at the GV stage. Fluorescence results showed that ALKBH5 signals were present primarily in the nucleus at the GV stage and then were uniformly dispersed in the oocytes at the MII stage (Supplementary Fig. 1b). Moreover, the protein levels of ALKBH5 were stable throughout meiotic maturation (Supplementary Fig. 1c).

To further examine the in vivo function of this factor, we generated *Alkbh5* total knockout mice by ablating exon 1, including the start codon, from the *Alkbh5* gene using the CRISPR-Cas9 system (Fig. 1a; Supplementary Fig. 2a). The successful knockout of ALKBH5 in oocytes was confirmed by western blot and IHC (Fig. 1b; Supplementary Fig. 2b). The resultant mice showed no developmental lethality but were sterile (Fig. 1c). As reported previously, *Alkbh5⁻/⁻* mice were smaller in size than their *Alkbh5⁺/⁺* littermates (Supplementary Fig. 2c). When heterozygous *Alkbh5⁺/⁻* mice were crossed in combination, we found that *Alkbh5⁻/⁻* mice were retrieved in sub-Mendelian ratios, with only 20% homozygous (less than the expected 25%) (Supplementary Fig. 2d). GV oocytes derived from PMSG-primed *Alkbh5⁻/⁻* females were fewer in number and had decreased diameters relative to those from control mice (Supplementary Fig. 2e, f). The nuclear configuration was also affected, as revealed by an increase in the ratio of non-surrounded nucleolus (NSN) to surrounded nucleolus (SN) oocytes (Supplementary Fig. 2g, h). Together, these results suggested impaired oocyte growth upon *Alkbh5* knockout.

Next, we super-ovulated 21-day-old mice and found that most *Alkbh5⁻/⁻* oocytes failed to release the first polar body (PB1) and harbored abnormal spindles, suggesting a key role of ALKBH5 in oocyte maturation (Supplementary Fig. 2i, j). However, we could not rule out nonspecific effects due to the lack of ALKBH5 in surrounding cells. Therefore, we cultured WT and *Alkbh5⁻/⁻* oocytes in vitro. Remarkably, most *Alkbh5⁻/⁻* oocytes were arrested at the GV stage and failed to undergo germinal vesicle breakdown (GVBD) (Fig. 1d, e). Fifteen hours after meiotic resumption, approximately 80% of WT oocytes but less than 10% of *Alkbh5⁻/⁻* oocytes had extruded PB1 (Fig. 1d, f). This was reminiscent of previous findings in vivo (Supplementary Fig. 2i, j), which confirmed the specific role of ALKBH5 in oocyte maturation.

Similar spindle assembly results were observed in vitro by immunofluorescence staining. During MI, 90% of control oocytes displayed a stable bipolar spindle with chromosomes aligned at the metaphase plate. In contrast, up to 30% of *Alkbh5⁻/⁻* oocytes failed to establish bipolar spindles, resulting in apolar and multipolar spindles with misaligned chromosomes (Fig. 1g–i). In addition, 40% of *Alkbh5⁻/⁻* oocytes, but none of the WT oocytes, had aggregated chromosomes at MI (Fig. 1i). These phenotypes were indicative of severe defects in microtubule organizing center (MTOC) clustering. Thus, we evaluated MTOC formation using antibodies against pericentrin and TPX2. In WT oocytes, pericentrin, a key component of the MTOC, was concentrated in a ring around the spindle poles (Fig. 1j). However, in *Alkbh5⁻/⁻* oocytes, we observed scattered pericentrin foci across the spindle body and decreased intensity of the microtubule marker TPX2 (Fig. 1j, k).

Intriguingly, despite the severe defects at MI, a small portion (4/49) of *Alkbh5⁻/⁻* oocytes succeeded in extruding PB1 at MII (Supplementary Fig. 2k). These PB1-released oocytes exhibited distorted spindles with abnormal chromatin clumps, indicating that they were unlikely to be competent to sustain further embryo development. To test this hypothesis, we super-ovulated *Alkbh5⁻/⁻* females at 4 weeks and mated them with WT males. After successful mating, fertilized oocytes were collected from the oviducts 28 h post hCG injection, and the fertilization rate was assessed by examining pronucleus formation. Notably, approximately half of the ovulated oocytes derived from *Alkbh5⁻/⁻* females were competent to be fertilized (Fig. 1l, m). To avoid interference from unfertilized oocytes, we next selected fertilized embryos for in vitro culture. Approximately 40% of zygotes reached the 2-cell stage, and none developed beyond the 2-cell stage (Fig. 1l, m), suggesting impaired MZT upon maternal *Alkbh5* knockout.

Collectively, these results indicated that ALKBH5 is necessary for oocyte growth and maturation, and affects further embryo development.

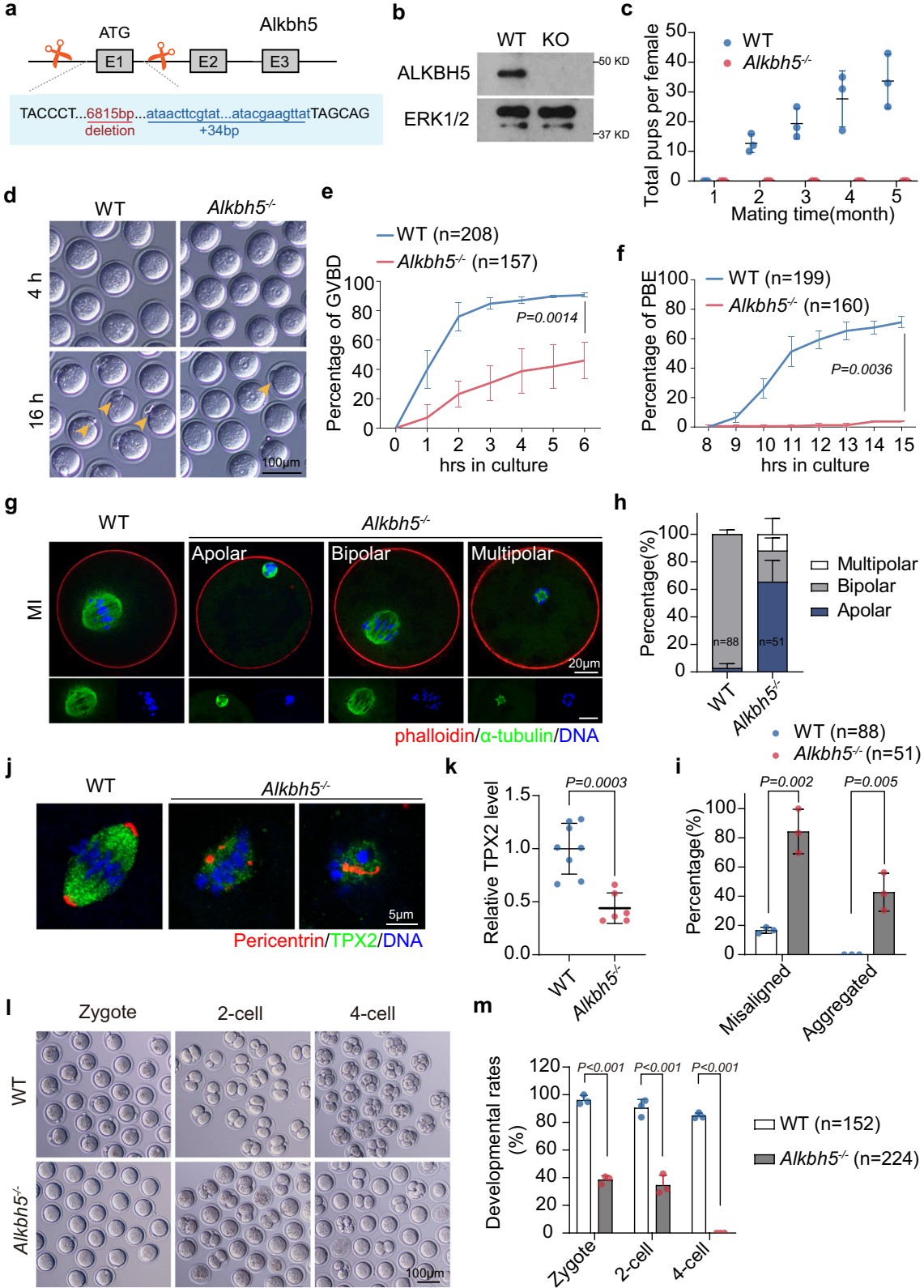

## Reduced MPF activity in the G2-M transition upon *Alkbh5* deletion

GVBD is driven by maturation-promoting factor (MPF), a dimeric protein complex composed of cyclin-dependent kinase 1 (CDK1) and Cyclin B1[22]. We therefore assessed the protein levels of Cyclin B1 and phosphorylated CDK1 at T161 (active CDK1). GV oocytes were released from milrinone to allow meiotic resumption. Oocytes that underwent GVBD were collected 4 h later and subjected to immunoblotting. Surprisingly, we found a substantial reduction in Cyclin B1 and a modest but appreciable downregulation of active CDK1 in *Alkbh5*[−/−] oocytes (Fig. 2a, b). In addition, the levels of Cyclin B2, another CDK1-activating cyclin, were comparable between WT and *Alkbh5*[−/−] oocytes (Fig. 2a, b). Cyclin B1 is targeted for degradation by the anaphase promoting complex/cyclosome (APC/C)-Cdh1 complex (APC/C[Cdh1]), and APC/C[Cdh1]

**Fig. 1 | *Alkbh5*$^{-/-}$ oocyte exhibits meiotic maturation defects. a** Scheme of *Alkbh5*-depletion strategy. **b** Immunoblot image comparing ALKBH5 in WT and *Alkbh5*$^{-/-}$ GV oocytes. ERK1/2 is applied as stable control. **c** Cumulative number of litter size derived from WT or *Alkbh5*$^{-/-}$ harem breeders over a 5-month period. *n* = 3 (WT), 3 (*Alkbh5*$^{-/-}$) female mice are used. Data are presented as mean ± SD. **d** Representative images of WT and *Alkbh5*$^{-/-}$ oocytes cultured in vitro at indicated time points. Cultured oocytes with polar body at 16 h are highlighted by arrowhead (yellow). **e**, **f** The rates of GVBD and PBE in WT and *Alkbh5*$^{-/-}$ oocytes respectively. The numbers of oocytes examined are indicated. Data are presented as mean ± SD. *P* value, two-tailed Student's *t* test. **g** Confocal microscopy of spindle assembly and chromosome alignment in MI (8 h) oocytes of WT and *Alkbh5*$^{-/-}$ mice cultured in vitro. Spindle labeled with α-Tubulin (green) and chromosome stained by DAPI (blue) are presented individually on the lower inset. Phalloidin marks the outliner of

oocyte in red. **h** The classification of multipolar, bipolar and apolar spindles in WT and *Alkbh5*$^{-/-}$ oocytes. The numbers of oocytes examined are indicated. Data are presented as mean ± SD. **i** The percents of misaligned and aggregated chromosomes in WT and *Alkbh5*$^{-/-}$ oocytes respectively. The data represent mean ± SD. *P* value, two-tailed Student's t test. **j** Magnified pictures resolving the immunostaining of MTOC in WT and *Alkbh5*$^{-/-}$ oocytes. **k** Quantification of TPX2 fluorescent intensity in (**j**). The data represent mean ± SD. *P* value, two-tailed Student's t test. *n* = 8 (WT), 6 (*Alkbh5*$^{-/-}$) oocytes are examined. **l** Representative pictures of embryos derived from WT and *Alkbh5*$^{-/-}$ oocytes at 28, 40 and 54 h after hCG injection. **m** Developmental rates of embryos that reach the indicated stages at corresponding timepoints. The data are presented as mean ± SD. *P* value, two-tailed Student's t test. *P* = 0.000023, 0.000513 and 0.000187 (from left to right). Source data are provided as a Source Data file.

activity is maintained by CDC14B[23]. However, western blot analysis of *Alkbh5*$^{-/-}$ oocytes showed unchanged levels of Cdh1 and CDC14B, indicating the existence of a noncanonical pathway for Cyclin B1 degradation after *Alkbh5* depletion (Supplementary Fig. 3a, b). Similar results were also detected in mouse granulosa cells in which endogenous *Alkbh5* expression was inhibited by the corresponding siRNA, which suggested that ALKBH5 might also be involved in mitosis (Supplementary Fig. 3c, d). Together, these results demonstrated that *Alkbh5* loss impairs MPF activity independent of the APC/C$^{Cdh1}$-Cyclin B1 pathway.

**MI arrest in *Alkbh5*$^{-/-}$ oocytes is due to inactivated APC$^{Cdc20}$**
Considering the MI arrest observed in *Alkbh5*$^{-/-}$ oocytes in vivo and in vitro (Supplementary Figs. 2i, 3e), we next asked whether homologous chromosomes remained conjoined. In WT oocytes, the expected 20 pairs of bivalents were well maintained before the first division, and 20 pairs of sister chromatids attached to one another were observed afterward. In contrast, chromosome spreads of *Alkbh5*$^{-/-}$ oocytes showed intact bivalent chromosomes after up to 16 h in culture, indicating a block of chromosome segregation (Fig. 2c). It was also evident that a moderate number of *Alkbh5*$^{-/-}$ chromosomes had relatively shorter arms or were fused with each other as fuzzy DNA signals at MI (Fig. 2c–e). These chaotic morphologies became more severe by MII, as disorganized chromosomes in combination with over-condensed DNA appeared (Fig. 2c, e). As a result of this impaired chromosome separation, we detected a higher rate of aneuploidy in *Alkbh5*$^{-/-}$ oocytes. Moreover, none of the *Alkbh5*$^{-/-}$ oocytes had 20 pairs of sister chromatids, while over 90% of the WT oocytes were euploid (Fig. 2f). These observations suggested that *Alkbh5* depletion inhibited homolog disjunction and caused excessive chromatin condensation, implying an unexpected role of ALKBH5 in chromosome organization.

APC is activated by the formation of the APC/C$^{Cdc20}$ complex, which enables the degradation of Securin and Cyclin B1, thus releasing separase activity[24]. The activated Separase then facilitates homologous chromosome segregation by cleaving arm Cohesin during the metaphase I (MI)-anaphase I (AI) transition[25]. Repressed Separase activity was indicated by unseparated chromosomes, as observed previously[26] (Fig. 2c). To determine the cause of Separase inactivation, we then examined the levels of the involved proteins at the AI via western blot. In contrast to the case in WT oocytes, Securin and Cyclin B1 both accumulated at anaphase onset in *Alkbh5*$^{-/-}$ oocytes, while the level of CDC20 decreased markedly (Fig. 2g, h). These results demonstrated APC/C$^{Cdc20}$ inactivation, resulting in the accumulation of its substrates.

Given that the restraint of Cyclin B1-mediated Separase relies in part on CDK1 activity, we next asked whether inhibiting CDK1 activity would enable *Alkbh5*$^{-/-}$ oocytes to undergo AI, despite the unchanged levels of CDK1-T161 phosphorylation (Fig. 2g, h). Indeed, the CDK1 activity inhibitor roscovitine partially but not completely rescued PB1 extrusion in *Alkbh5*$^{-/-}$ oocytes (Fig. 2i, j). Notably, some PB1-extruded *Alkbh5*$^{-/-}$ oocytes with an interphase-like nucleus were detected after

roscovitine treatment, recapitulating the phenotype of GDF9-*Ccnb1*$^{-/-}$ oocytes[27] (Fig. 2i, k). In summary, we concluded that inactivated APC/C$^{Cdc20}$ contributes to MI arrest by promoting the accumulation of Securin and Cyclin B1. Inhibiting Cyclin B1-CDK1 activity could not fully rescue *Alkbh5*$^{-/-}$ oocytes, indicating that high levels of Securin may be the main cause of this phenotype.

**Inhibition of Mps could not fully restore progression to MII**
Previous studies have shown that satisfaction of the spindle assembly checkpoint (SAC) allows for timely APC/C$^{Cdc20}$ activation and that inappropriate SAC activity induces Securin accumulation[28]. We hence speculated that SAC prevented AI in *Alkbh5*$^{-/-}$ oocytes. In support of this hypothesis, immunofluorescence analysis revealed prolonged SAC activation upon *Alkbh5* knockout. In contrast to the case in WT oocytes, BubR1, a key component of the SAC, was recruited to kinetochores before MI but failed to leave at AI in *Alkbh5*$^{-/-}$ oocytes (Fig. 3a, b; Supplementary Fig. 3f, g). Thus, we inhibited the SAC kinase Mps by adding reversine to the culture medium before AI onset (Fig. 3c). As expected, PBE was accelerated in both WT and *Alkbh5*$^{-/-}$ oocytes. However, to our surprise, PBE in *Alkbh5*$^{-/-}$ oocytes was only partially recovered after reversine treatment and was not fully restored to control values (Fig. 3d). Closer analysis of AI-related proteins confirmed overactivation of APC/C$^{Cdc20}$ in reversine-treated *Alkbh5*$^{-/-}$ oocytes, which indicated that *Alkbh5*$^{-/-}$ oocytes had already entered the AI (Fig. 3e, f). Chromosome compaction has been found in some cases of chromosome separation failure, with chromosomes unable to be pulled by microtubules[29,30]. Hence, we suspected that over-condensed chromosomes may affect the localization of Separase, thus resulting in PBE failure even after bypass of the SAC. In support of this idea, approximately 40% of *Alkbh5*$^{-/-}$ oocytes showed ball-shaped chromosomes without spindle formation (Fig. 3g).

Taken together, these results indicated that failure to reach MII in *Alkbh5*$^{-/-}$ oocytes was partially attributable to SAC activation and that chromosome compaction caused by *Alkbh5* loss might affect Separase activity (Fig. 3h).

**Knockout of *Alkbh5* impacts the decay of maternal transcripts**
m$^6$A modification serves as a key determinant in posttranscriptional RNA regulation and contributes to a wide range of RNA outcomes, including RNA maintenance, degradation, and transport[7]. To investigate how the m$^6$A eraser ALKBH5 affects the maternal transcriptome, we subjected oocytes at the GV, GVBD and MII stages (after 0, 4, and 16 h of in vitro culture) to RNA-seq with an exogenous ERCC RNA mix as a spike-in control (Fig. 4a). Correlation analysis showed similar expression patterns between two replicates of the same stage (Supplementary Fig. 4a). As expected, we observed completely different transcriptomes in the absence of ALKBH5, as revealed by heatmap and PCA clustering (Fig. 4b, c). Efficient *Alkbh5* knockdown at the gene level was validated, and *Actb* was applied as an internal control in the following experiments, as its level remains stable throughout meiotic maturation (Fig. 4d).

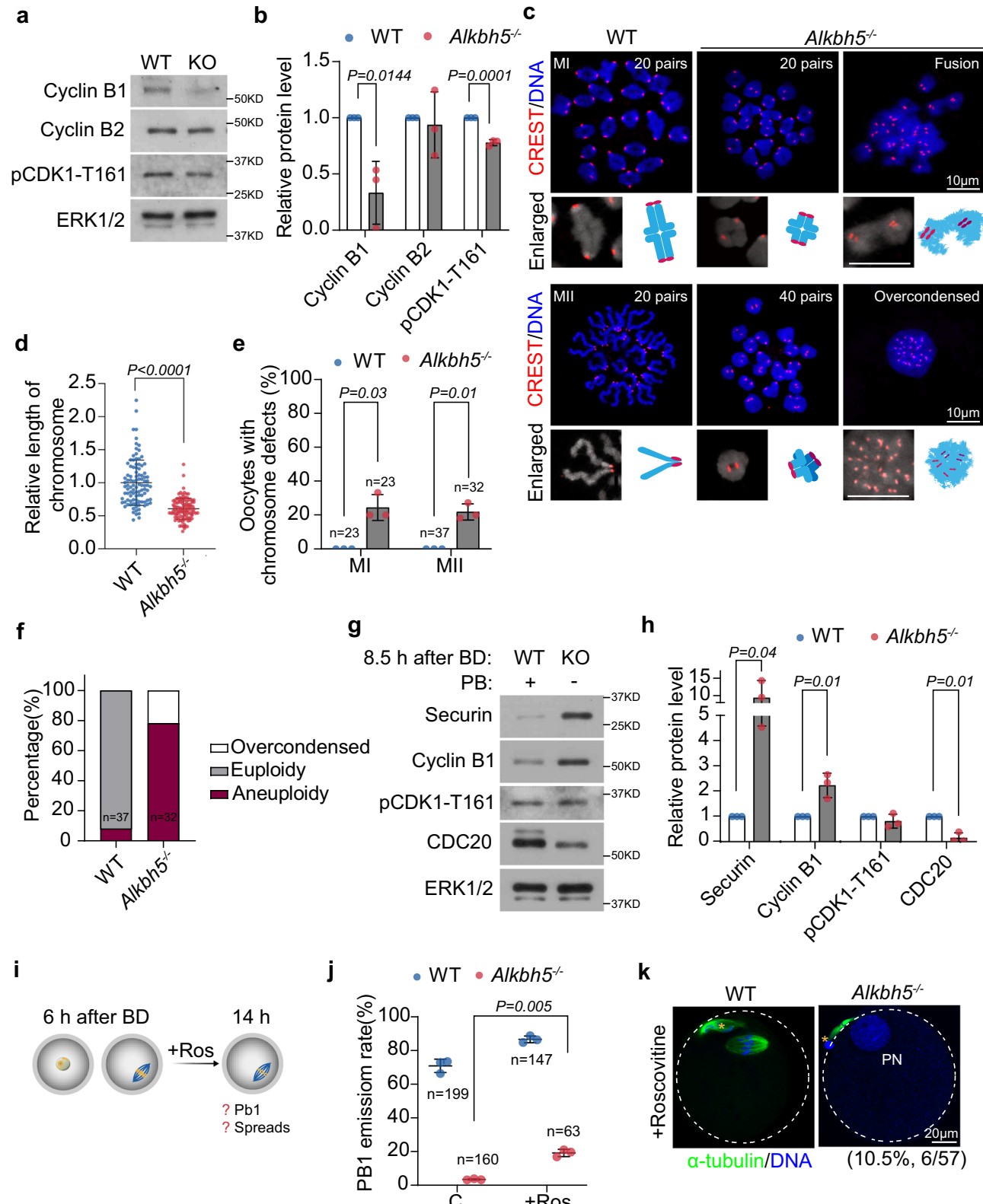

Next, we compared library size between WT and *Alkbh5^{-/-}* samples with ERCC normalization. In line with previous findings[4,31], approximately half of the transcript pool was degraded during the GV-MII transition in WT oocytes. However, oocytes depleted of *Alkbh5* showed a mild delay in RNA clearance (Fig. 4e). Detailed comparative analysis supported the finding of RNA accumulation: a nearly normal transcriptome was observed in *Alkbh5*-null GV oocytes, with minor differences (179 transcripts with reduced abundance, 244 with increased abundance). These differences became more pronounced by the BD stage (462 transcripts with reduced abundance, 447 with increased abundance), and transcripts with increased abundance outnumbered transcripts with decreased abundance at the MII stage (238 transcripts with reduced abundance, 706 with increased abundance) (Fig. 4f, g; Supplementary Data 1–3). These data suggested that

**Fig. 2 | ALKBH5 is required for meiotic cycle progression.** Representative images (**a**) and quantifications (**b**) of meiotic-resumption related protein levels (Cyclin B1, Cyclin B2, p-CDK1 (T161)) in GVBD oocytes. ERK1/2 is referred as a loading control. The bars represent mean ± SD. *P* value, two-tailed Student's t test. *n* = 3 biologically independent experiments. **c** Chromosome spreads revealing homologous chromosome disjunction and sister chromatid separation in MI and MII oocytes respectively. The enlarged individual chromosome is presented on the left inset with a model of which on the right. **d** The relative length of chromosome arms in MI oocytes. Each dot represents the length of chromosome arm from an individual oocyte. Data are expressed as mean ± SD. *P* value, two-tailed Student's t test. *P* = 3.03e−19. *n* = 100 (WT), 106 (*Alkbh5*⁻/⁻) samples are examined. **e** The percents of oocytes with chromosome defects in MI and MII respectively. Data represent the mean ± SD from three replicates and numbers of examined oocytes are shown. *P* value, two-tailed Student's t test. **f** Stacked bar chart showing the ratio of euploidy

(20 pairs), aneuploidy (>20 or <20 pairs) and over-condensed chromatin (pair number undefined) in WT and *Alkbh5*⁻/⁻ oocytes. Note the absence of euploidy in *Alkbh5*⁻/⁻ oocytes. Representative images (**g**) and quantifications (**h**) of MI-AI associated protein levels (Securin, Cyclin B1, p-CDK1 (T161) and CDC20) 8.5 h after onset of GVBD. ERK1/2 is presented as an internal control. The bars represent mean ± SD. *P* value, two-tailed Student's t test. *n* = 3 biologically independent experiments. **i** Schematic illustration of roscovitine (CDK1 inhibitor) treatment on oocytes at indicated timepoints. Ros, roscovitine. **j** The PB1 emission rates of WT and *Alkbh5*⁻/⁻ oocytes with or without roscovitine treatment. Data are depicted as mean ± SD of three replicates and counts of examined oocytes are indicated. *P* value, two-tailed Student's t test. **k** Fluorescence imaging of oocytes cultured at indicated timepoint after roscovitine treatment. Extruded polar bodies are denoted by asterisk (yellow). PN pronucleus. Source data are provided as a Source Data file.

ALKBH5 may have a slight impact on the formation of the maternal transcriptome but extensively regulates transcript dosage after meiotic resumption. In addition, a quadrant diagram revealed that among 706 transcripts differentially upregulated at the MII stage, 649 showed dramatic degradation across maturation, while only 1 transcript was stabilized (Fig. 4h). In summary, the loss of ALKBH5 enhances the abundance of a cohort of transcripts that should be destabilized during the GV-MII transition.

Analysis of functional enrichment data also showed that a variety of fundamental biological processes are involved at each stage: peptidase and endopeptidase activity at the GV stage and T-cell activation at the GVBD stage (Supplementary Fig. 4b). Notably, pathways associated with translation and oxidative phosphorylation were significantly enriched at the MII stage (Fig. 4i; Supplementary Fig. 4b), suggesting that mRNAs encoding ribosomes and ATP synthesis may be subjected to ALKBH5 control.

### *Alkbh5* loss disturbs RNA stability and causes excessive translation

To identify the class of genes regulated by ALKBH5, we sorted maternal transcripts across oocyte maturation into two groups: stabilized genes (MII/GV > 2, *P* < 0.05) and destabilized genes (MII/GV < 0.5, *P* < 0.05) (Supplementary Data 4). Within the WT transcriptome, 2929 transcripts exhibited were markedly decreased abundance from GV to MII, while only 60 transcripts were stabilized (Fig. 5a). The bias toward destabilization is consistent with the fact that the degradation of maternal RNAs is coupled with the cessation of transcription. Nevertheless, only 1810 transcripts were found to be degraded during the GV-MII transition in *Alkbh5* KO oocytes (Fig. 5a). Additionally, a subset of transcripts (1529) were degraded exclusively in WT and stabilized upon *Alkbh5* knockout (Fig. 5b; Supplementary Data 4). We also noticed that *Alkbh5* loss depressed the degradation of transcripts (1400) that showed decay in both WT and KO oocytes (Supplementary Fig. 4c, d), suggesting the global destabilizing role of ALKBH5 in the overall transcriptome. These results further supported the idea that *Alkbh5* deficiency prevents RNA decay for certain subsets of transcripts. Subsequent GO and KEGG analyses revealed that those 1529 transcripts are enriched for translation-related and mitochondria-related processes, such as ribosome and ATP synthesis (Fig. 5c; Supplementary Fig. 4e). Thus, for the following experiments, we selected several genes from this subset that are implicated in ribosome functions (*Rpl39*, *Rpl23a*, *Rpl5*, *Rps13*), ATP synthesis (*Atp5j2*), and oocyte growth (*Uchl1*, *Birc5*, *Esrrb*) as representatives. The impaired degradation of these transcripts in *Alkbh5* KO oocytes was validated by RT-qPCR (Fig. 5d).

In cytoplasmic RNA metabolism, polyadenylation and deadenylation control poly(A) tail length, which in turn determines mRNA fate[32]. To directly evaluate deadenylation activity, total RNA was isolated from oocytes and reverse-transcribed with oligo anchor probes for PAT analysis. As reported previously[4], the poly(A) tails of maternal

transcripts were shortened during oocyte maturation in WT oocytes. However, this process was compromised after *Alkbh5* depletion, which could account for the RNA stabilization triggered by *Alkbh5* loss (Fig. 5e). In support of this, we also observed upregulated expression of polyadenylation-related genes from the RNA-seq dataset (Supplementary Fig. 4f).

Since accumulated RNAs were enriched for translation, we next addressed whether global translation activity was affected. L-homopropargylglycine (HPG), an indicator of newly synthesized proteins, was utilized to detect translation activity. We found that the HPG signals in *Alkbh5* KO oocytes were much higher than those of control oocytes (Fig. 5f, g). In addition, some *Alkbh5*-null oocytes without polar bodies contained aggregated HPG foci, indicating abnormal translation (Fig. 5f).

To summarize, these results suggest that ALKBH5 is crucial for maternal RNA decay and that the loss of ALKBH5 contributes to excessive translation.

### The m⁶A methylome is remodeled in GV oocytes depleted of ALKBH5

ALKBH5 catalyzes m⁶A demethylation activity to remove the methyl group from modified RNAs. To capture the m⁶A dynamics in the absence of ALKBH5, we performed m⁶A-methylated RNA immunoprecipitation sequencing (MeRIP-seq) on approximately 2800 GV oocytes from WT and *Alkbh5*⁻/⁻ female mice (Fig. 6a).

We found that m⁶A was confined to the consensus motif "RRACH" (R = G or A; H = A, C, or U) (Fig. 6b) and preferentially resided in the 3′ UTR and near the stop codon of the exon (Fig. 6c; Supplementary Fig. 5a), a result that was reminiscent of other previously published research[16,17]. Notably, the *Alkbh5* KO sample showed slightly higher m⁶A enrichment near the stop codon, suggesting a modest m⁶A enhancement after *Alkbh5* knockout (Fig. 6c). The density distributions of the peak length and peak number of modified transcripts were almost comparable between WT and *Alkbh5*⁻/⁻ oocytes (Supplementary Fig. 5b, c). A closer comparative analysis of the m⁶A-seq dataset revealed that a total of 2450 m⁶A peaks were differentially changed upon *Alkbh5* knockout, with 1325 upregulated and 1125 downregulated enrichments (Fig. 6d). The unexpectedly high proportion (1125 out of 2450) of hypomethylated peaks observed in *Alkbh5* KO oocytes is presumably due to the limited impact of ALKBH5 on GV oocytes, in which ALKBH5 exhibits nuclear localization. The transcripts with changed m⁶A marks were functionally enriched in fundamental biological processes including DNA repair, cell cycle and chromatin organization (Fig. 6f); for example, the chromatin organization-associated gene *Sted6* was identified to contain hypermethylated peaks after *Alkbh5* knockout (Fig. 6e). Taken together, these data indicated a remodeled m⁶A landscape in *Alkbh5*⁻/⁻ oocytes.

Previous studies have demonstrated that m⁶A regulates maternal RNA degradation in a timely fashion during the maternal-to-zygotic transition (MZT) and controls the translation and

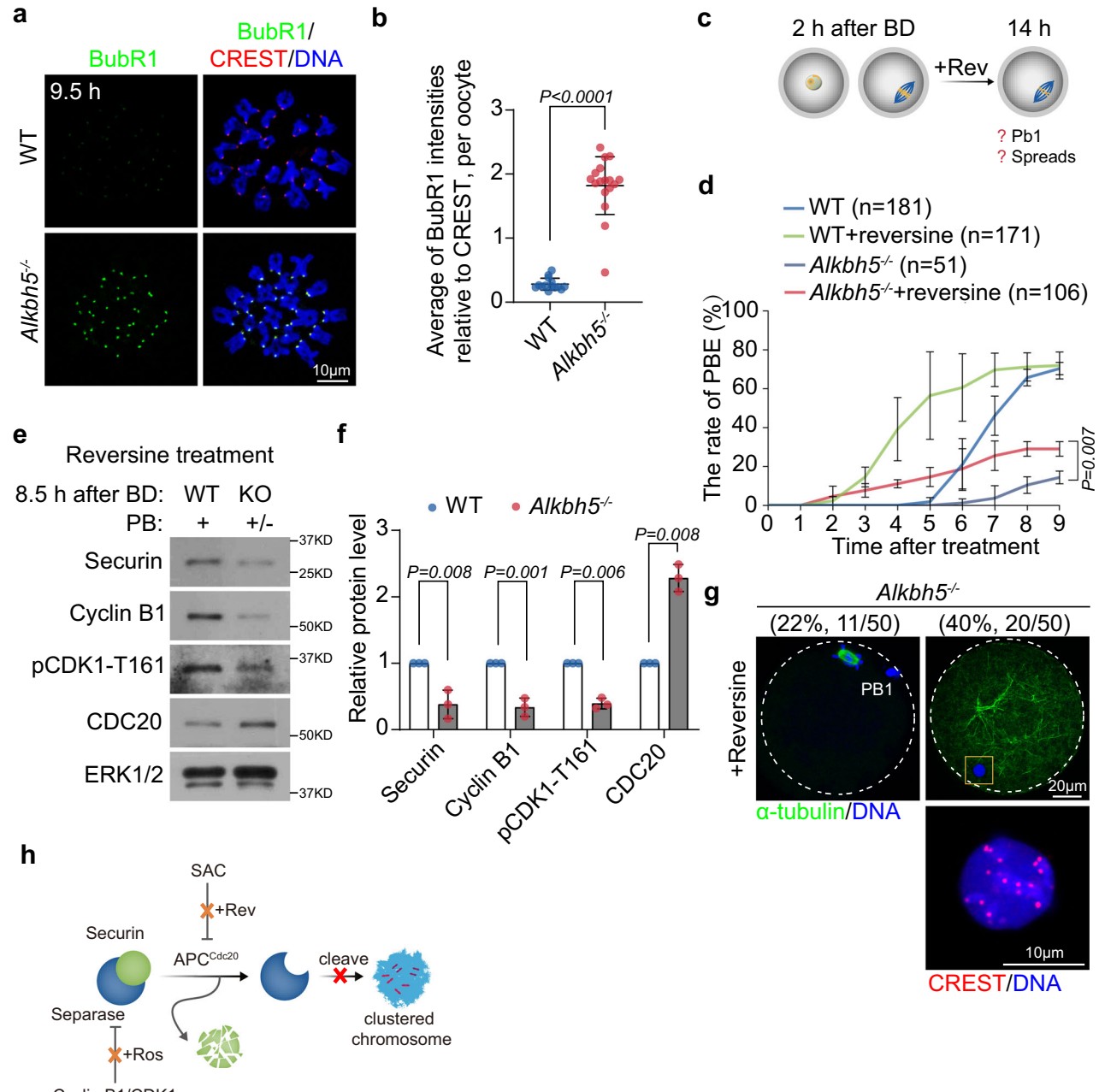

**Fig. 3 | MI arrest in *ALKBH5*⁻/⁻ oocyte is partially due to the prolonged SAC activation. a** Immunostaining showing BubR1 staining at kinetochores from WT and *Alkbh5*⁻/⁻ oocytes at anaphase I (9.5 h). **b** Quantifications of BubR1 intensity in (**a**). BubR1/ Crest is expressed as the relative intensity compared to the controls. The measurement of average BubR1 signal is detailed in "Methods". Data are shown in the form of mean ± SD and each dot represents an individual oocyte. *P* value, two-tailed Student's t test. *P* = 3.33e−13. *n* = 14 (WT), 17 (*Alkbh5*⁻/⁻) biologically independent oocytes are examined. **c** Scheme of reversine (Mps inhibitor) treatment on oocytes at indicated timepoints. Rev reversine. **d** The PB1 emission rates of WT and *Alkbh5*⁻/⁻ oocytes at indicated timepoints with or without reversine treatment. Numbers of oocytes applied are depicted. Data are presented as mean ± SD. *P* value,

two-tailed Student's t test. Representative images (**e**) and quantifications (**f**) of MI-AI associated protein levels (Securin, Cyclin B1, p-CDK1 (T161) and CDC20) after reversine treatment. ERK1/2 is presented as an internal control. The bars represent mean ± SD. *P* value, two-tailed Student's t test. *n* = 3 biologically independent experiments. **g** Spindle and chromosome morphologies of WT and *Alkbh5*⁻/⁻ oocytes after reversine treatment. The overcondensed chromosome is indicated and magnified on the lower inset. The outliners of oocytes are marked by white dashed circle. PB1, first polar body. **h** Schematic illustration of mechanism regarding unseparated meiotic bivalents in *Alkbh5*⁻/⁻ oocytes after addition of SAC inhibitors reversine. Source data are provided as a Source Data file.

stability of certain m⁶A-modified RNAs[18,33]. Since *Alkbh5*⁻/⁻ oocytes were proven to exhibit delayed RNA clearance, accompanied by globally elevated translation, we next aimed to determine whether m⁶A is involved in this process. As reported previously[16,17], we found that transcripts lacking m⁶A are degraded more rapidly than m⁶A-modified RNAs during the GV-to-MII transition (Fig. 6g), which confirms the previously advanced notion that m⁶A protects

maternal RNAs before fertilization. Compared with the controls, *Alkbh5* depletion led to a global delay in RNA decay for both m⁶A-modified and unmodified RNAs, with m⁶A-tagged transcripts displaying the slowest decay rate (Fig. 6h, i), resulting in a greater abundance of transcripts with m⁶A modification at the MII stage (Fig. 6j). Therefore, m⁶A appears to be associated with ALKBH5-mediated decay failure.

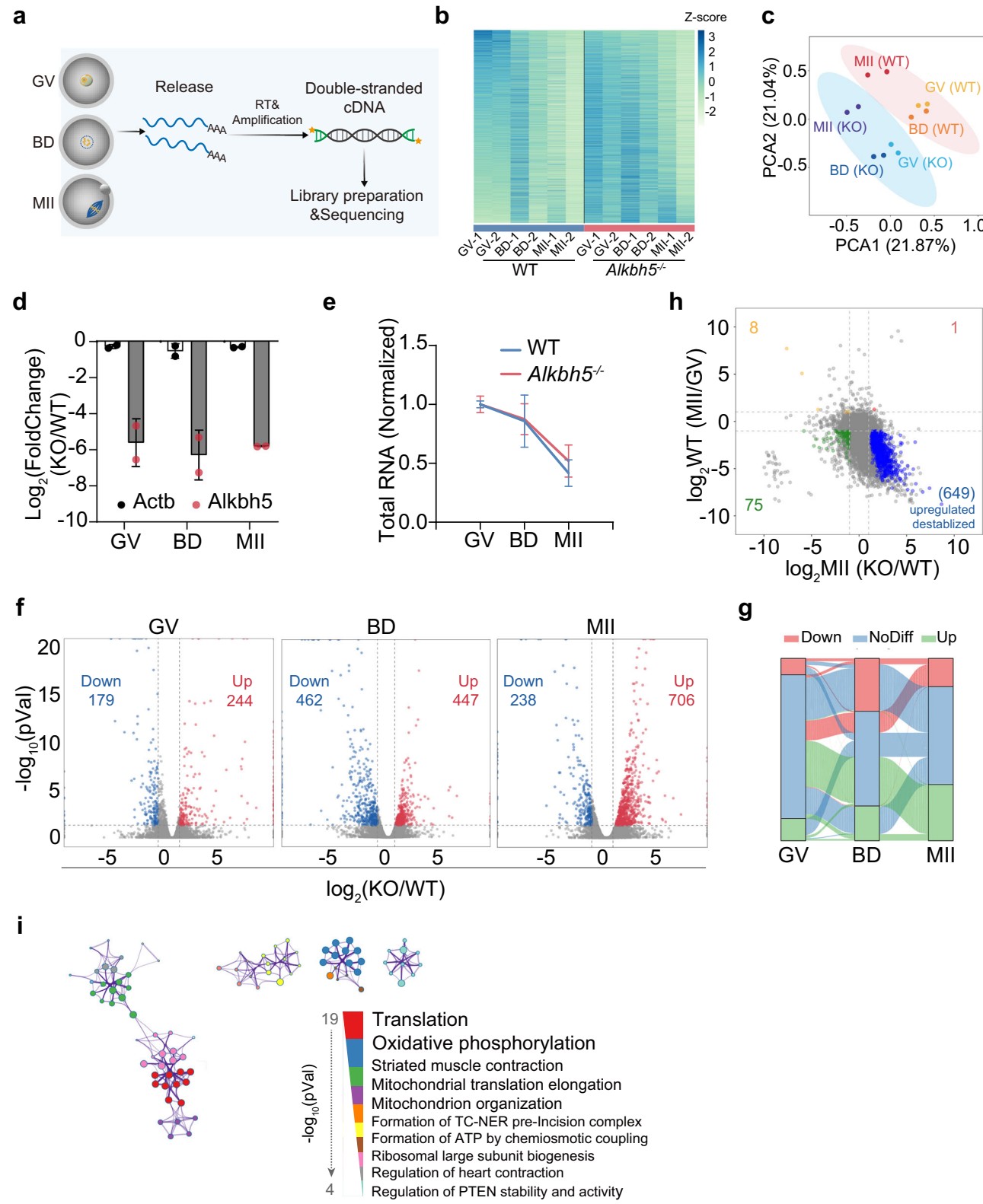

Very recently, a similar m⁶A profile of mouse oocytes and embryos was reported using low-input methyl RNA immunoprecipitation and sequencing (picoMeRIP-seq)[18]. Meanwhile, the m⁶A landscape of a single oocyte during maturation was well characterized by single-cell m⁶A sequencing (scm⁶A-seq)[16]. We therefore compared our profile with that in these studies and found that nearly 86% of m⁶A-tagged transcripts detected by picoMeRIP-seq and approximately 68% by

scm⁶A-seq were identified in GV oocytes within our dataset using bulk MeRIP-seq (Supplementary Fig. 5d). In agreement with the existing findings, the 2110 transcripts defined as modified by all three different mapping techniques were greatly involved in the regulation of transcription, chromatin organization and the cell cycle (Supplementary Fig. 5e, f). Herein, we designated a gene as m⁶A⁺ if it was identified in both our dataset and at least one of the other two published datasets

**Fig. 4 | Altered transcriptional profiles in *Alkbh5*$^{-/-}$ oocytes. a** Flow chart of SMART-seq. 20 GV, BD and MII oocytes from WT and *Alkbh5*$^{-/-}$ females are collected at 0, 4 and 16 h respectively, and subjected to sequencing as detailed in "Methods". Two biological independent samples are reported. **b** Heatmap showing gene expressions in WT and *Alkbh5*$^{-/-}$ oocytes. Each row represents one gene and each column represents one sample. FPKM of each gene are normalized to Z-score indicated by color scale. **c** PCA clustering of sequenced samples. Each dot represents an individual library, labeled with genotype and development stage. **d** The relative expression levels of *Alkbh5* and *Atcb* in GV, BD and MII oocytes. $n = 2$ biological replicates. Data are presented as mean ± SD. **e** Comparison of total RNA levels in WT and *Alkbh5*$^{-/-}$ SMART-seq libraries at each indicated stage. $n = 2$ biological replicates. Data are presented as mean ± SD. **f** Volcano plots showing differentially expressed genes (DEGs) between WT and *Alkbh5*$^{-/-}$ groups at the GV, BD and MII stages. Each dot represents an individual gene and numbers of DEGs are

indicated. Genes differential expression analysis was performed by DESeq2 software between two different groups. DEGs cut-off is false discovery rate (FDR) < 0.05 and absolute fold change≥2. BD, GVBD. **g** Sankey diagram depicting the dynamic composition of significantly up-regulated, no-differential, or down-regulated transcripts from the GV to MII stages. **h** Four quadrant diagram showing correlation between maternal transcriptome from the GV to MII stages and gene expressions in *Alkbh5*$^{-/-}$ MII oocytes with respect to WT MII oocytes. The significantly upregulated genes in *Alkbh5*$^{-/-}$ MII oocytes that are degraded in GV-MII transition are highlighted in blue. The number of genes in each gate is shown. **i** Network plot presenting functional enrichment analysis of DEGs at MII stage. Metascape was applied as detailed in "Methods". The *P* values are calculated using hypergeometric test of functional annotation with default parameters. Top 10 ranked annotations are reported and their *P* values are shown. Source data are provided as a Source Data file.

and applied them for the same analyses conducted above. Similar results were obtained (Supplementary Fig. 5g–j). Collectively, these results validated the reproducibility and quality of our m$^6$A-seq dataset.

### ALKBH5-mediated m$^6$A removal destabilizes loss-GMD RNAs in oocytes

The above results led us to hypothesize that ALKBH5, as an integral m$^6$A eraser, may contribute to timely RNA decay through the removal of m$^6$A. To uncover the specific substrates of ALKBH5 in these contexts, we identified a subpopulation of transcripts as GV-to-MII decayed RNAs (abbreviated as "GMD" hereafter) on the basis of the RNA-seq dataset (Supplementary Data 5). The levels of these 2929 RNAs decreased remarkably (fold change < 0.5, $P < 0.05$) during the GV-to-MII transition. Since the requirement of large amounts of material for m$^6$A profiling hindered our ability to perform this analysis on MII oocytes, we combined our RNA-seq datasets of GV and MII oocytes derived from *Alkbh5*-knockout and wild-type mice with a published profile of m$^6$A dynamics from GV to MII stages[18].

Among 2929 GMD transcripts, 855 transcripts harbored distinct m$^6$A peaks ($P < 0.05$) at the GV stage, which were designated m$^6$A$^+$ GMD, while the remaining 2074 transcripts were named m$^6$A$^-$ GMD (Fig. 7a). The degradation of both m$^6$A$^+$ and m$^6$A$^-$ GMD transcripts was repressed in *Alkbh5*-deficient oocytes, similar to the transcriptome changes in m$^6$A-tagged and untagged RNAs that occur after *Alkbh5* knockout (Fig. 7b). Of note, GMD transcripts marked by m$^6$A displayed the most pronounced degradation dysfunction. We also detected elevated expression levels at the MII stage upon *Alkbh5* depletion, with m$^6$A$^+$ GMD showing the highest RNA abundance (Fig. 7b). These data imply that m$^6$A$^+$ GMD transcripts are likely regulated by ALKBH5.

A notable loss of m$^6$A$^+$ transcripts was observed between the GV and MII stages, suggesting the occurrence of demethylation activity catalyzed by ALKBH5 during this process (Fig. 7a). Detailed calculation further revealed that 298 transcripts (termed "loss-GMD") lost their m$^6$A modification once they reached the MII stage, while 557 transcripts (termed "inherited-GMD") retained their m$^6$A enrichment throughout oocyte maturation (Fig. 7a). Of these, loss-GMD transcripts emerged as potential targets of ALKBH5 in mouse oocytes after the GV stage.

The proportion of transcripts that lost m$^6$A in m$^6$A-marked GMD was significantly higher than that in m$^6$A-marked non-GMD transcripts (Fig. 7c). Further analysis revealed an accelerated decay of loss-GMD transcripts relative to that of their inherited-GMD counterparts (Fig. 7d). This evidence highlights the critical role of m$^6$A loss in RNA degradation, indicating that m$^6$A is preferentially lost in decayed transcripts and that transcripts that have lost m$^6$A tend to be less stabilized. However, loss-GMD transcripts seemed to maintain their methylation at the MII stage in *Alkbh5* KO oocytes, as the difference in degradation between loss- and inherited-GMD became less obvious

(Fig. 7d). In support of this idea, we found that m$^6$A enrichment of candidate transcripts in this group (*Atp5j2, Rpl39, Birc5, and Esrrb*) was drastically decreased by the MII stage in WT oocytes. Nevertheless, this phenotype was much less pronounced upon *Alkbh5* knockout, and these transcripts possessed significantly stronger m$^6$A enrichment at the MII stage (Fig. 7e–h). Therefore, *Alkbh5* deficiency causes gene-selective maintenance of m$^6$A in MII oocytes.

In addition to maintaining m$^6$A status at the MII stage, endogenous substrates were also expected to be stabilized in the absence of ALKBH5 (Fig. 7i). To test this hypothesis, we then compared degradation patterns between WT and KO samples. Indeed, both the rate and the extent of degradation in KO oocytes were much lower than those in the controls (Fig. 7j). A closer examination revealed that most loss-GMD transcripts resisted degradation: among 298 transcripts that had lost m$^6$A by the MII stage, 237 (79.5%) showed impaired RNA clearance (KO (MII/GV)/WT(MII/GV) > 1); of these, 135 (67.0%) remained unchanged or even increased in abundance in KO oocytes (Fig. 7k). Consistently, 94.3% (50/53) of the changed loss-GMD transcripts exhibited increased abundance at the MII stage after *Alkbh5* knockout. In contrast, only a limited proportion (3/53) exhibited lower abundance (Fig. 7l).

We next validated the binding of these transcripts by ALKBH5 in mouse oocytes using RIP-qPCR (Fig. 7m–p). Additionally, the effect of ALKBH5 on those candidate transcripts was recapitulated in granulosa cells derived from PMSG-primed *Alkbh5*$^{-/-}$ mice, supporting these RNAs as ALKBH5 substrates in most tissues (Supplementary Fig. 6a, b). GO analyses revealed that loss- and inherited-GMDs were functionally distinct. The loss-GMD genes were greatly enriched in translation (Supplementary Fig. 6c). As demonstrated above, these translation changes were largely recapitulated in *Alkbh5* KO oocytes, with MII oocytes exhibiting an accumulation of HPG signals (Fig. 5f, g). Taken together, these data suggest that ALKBH5 removes m$^6$A from loss-GMD transcripts, thereby ensuring timely RNA decay during the GV-to-MII transition.

### ALKBH5 function also precedes the onset of maternal RNA decay

Since *Alkbh5* depletion also led to compromised decay in m$^6$A-unmarked GMD, we cannot rule out the possibility that m$^6$A$^-$ transcripts lose their m$^6$A due to ALKBH5 activity at a very early phase of oocyte growth, which results in their subsequent degradation during the GV-MII transition (Supplementary Fig. 6e). To this end, we grouped m$^6$A-modified transcripts into two clusters according to the existence of ALKBH5. A total of 573 transcripts were defined as ALKBH5-dependent m$^6$A-modified (A5$^+$m$^6$A$^+$) transcripts, which lost m$^6$A peaks in WT oocytes but gained m$^6$A exclusively in KO oocytes (Supplementary Fig. 6f; Supplementary Data 6). Among them, only 136 transcripts were degraded markedly from the GV to MII stage in WT (Supplementary Fig. 6g), as exemplified by an IGV snapshot of a representative gene, Spc*24* (component of the Ndc80 complex) (Supplementary Fig. 6j). A

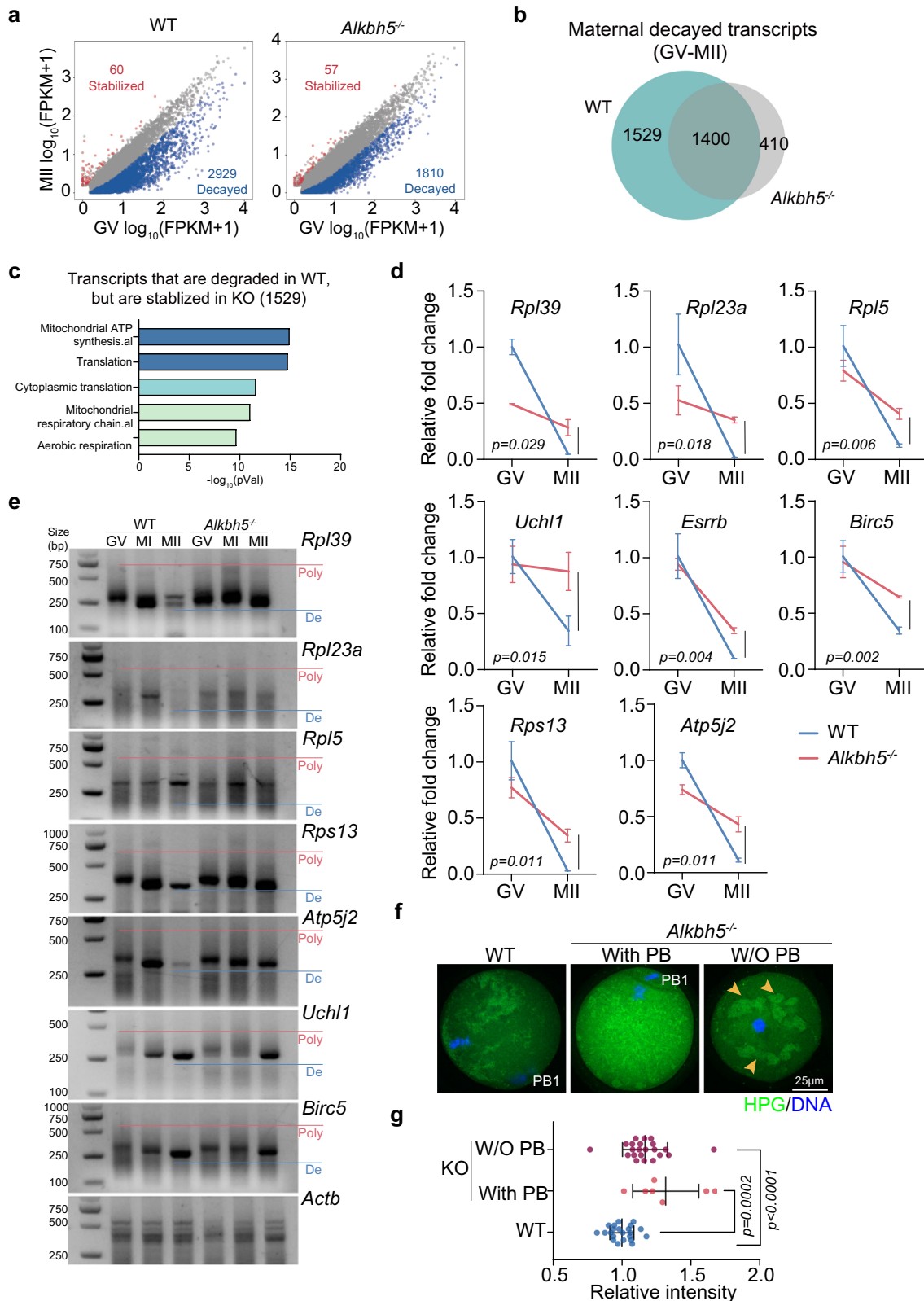

**a** WT / Alkbh5⁻/⁻
60 Stabilized / 2929 Decayed
57 Stabilized / 1810 Decayed

**b** Maternal decayed transcripts (GV-MII)
WT 1529 / 1400 / 410 Alkbh5⁻/⁻

**c** Transcripts that are degraded in WT, but are stablized in KO (1529)

**d** Rpl39 p=0.029, Rpl23a p=0.018, Rpl5 p=0.006, Uchl1 p=0.015, Esrrb p=0.004, Birc5 p=0.002, Rps13 p=0.011, Atp5j2 p=0.011 — WT / Alkbh5⁻/⁻

**e** WT / Alkbh5⁻/⁻ — Rpl39, Rpl23a, Rpl5, Rps13, Atp5j2, Uchl1, Birc5, Actb

**f** WT / Alkbh5⁻/⁻ With PB / W/O PB — HPG/DNA

**g** KO W/O PB, With PB, WT — p=0.0002, p<0.0001

similar but milder degradation pattern and extent of degradation were observed in this subset of transcripts (Supplementary Fig. 6h, i). Intriguingly, A5⁺m⁶A⁺ GMD genes and loss-GMD genes seemed to mediate similar functions, as both sets were enriched in translation pathways (Supplementary Fig. 6k). These results indicated that demethylation failure before the GV stage only accounts for a portion of the defective RNA clearance caused by *Alkbh5* depletion.

## IGF2BP2 is implicated in maturation defects caused by *Alkbh5* loss

Having demonstrated that persistent m⁶A modification caused aberrant RNA decay, we next wondered which m⁶A reader is involved in this process. There are numerous m⁶A readers, and some of them have been proven to be indispensable regulators of RNA stability, including YTHDF2, YTHDC1 and IGF2BPs[10,13,17,33,34]. To determine the specific m⁶A

**Fig. 5 | Alkbh5 loss induces RNA accumulation and a consequent over-translation in oocytes. a** Scatterplots comparing the relative expression changes from the GV to MII stages in WT and *Alkbh5*<sup>−/−</sup> samples respectively. Significantly up- and downregulated genes are highlighted (||log₂[fold change (MII/GV)]|| > 1, *q* < 0.05). **b** Venn diagram of maternal decayed genes (fold change (MII/GV) < 0.5, *q* < 0.05) compared between WT and *Alkbh5*<sup>−/−</sup> oocytes. **c** GO analysis of maternal transcripts that are degraded in WT oocytes, but are stabilized in *Alkbh5*<sup>−/−</sup> oocytes, as indicated in (**b**). The *P* values are calculated using hypergeometric test of functional annotation in DAVID website with default parameters. The top 5 categories are reported. **d** Comparisons between degradation rates of representative RNAs in WT and *Alkbh5*<sup>−/−</sup> oocytes during GV-MII transition. The mRNA levels are normalized to *Actb*. The data are presented as mean ± SD. *P* value, two-tailed Student's t test.

*n* = 3 biological replicates. **e** PAT analysis showing the changes in poly(A)-tail length in WT and *Alkbh5*<sup>−/−</sup> oocytes during meiotic maturation. Poly, polyadenylated; De deadenylated. **f** Representative images of global translation activity in WT and *Alkbh5*<sup>−/−</sup> oocytes by culture in the presence of HPG. The abnormal aggregated foci are indicated (right panel; yellow arrowhead). Extruded polar bodies are highlighted. W/O, without; PB1, first polar body. **g** Quantifications of HPG intensity in (**f**). An average fluorescence intensity is estimated by single oocyte measurement and is plotted as a single dot. The mean and SD are from three or more repeated recordings. *P* value, two-tailed Student's t test. W/O PB vs. WT, *P* = 0.000115. *n* = 22 (WT), 7 (KO with PB) and 22 (KO without PB) oocytes are examined. Source data are provided as a Source Data file.

reader, we analyzed the binding sites of m⁶A readers (YTHDF2, YTHDC1, IGF2BP2 and IGF2BP3) in public eCLIP-seq and RIP-seq datasets derived from human and mouse embryonic stem cells[35–37]. Compared with those of other readers, IGF2BP2 binding sites showed the most significant overlap with the loss-GMD m⁶A peaks (Fig. 8a). Consistently, the enrichment of IGF2BP2 binding for targeted transcripts (*Atp5j2*, *Rpl39*, *Birc5*, and *Esrrb*) was markedly higher than that of any other readers in GV oocytes (Fig. 8b–e). RIP-qPCR in mGC further confirmed the target specificity of IGF2BP2 (Supplementary Fig. 6d). Most importantly, we found that the binding of IGF2BP2 to these transcripts was greatly disrupted in WT oocytes once they reached the MII stage. In contrast, strong IGF2BP2 enrichment at these transcripts was still observed in KO oocytes at the MII stage (Fig. 8f–i). This is in line with the previous notion that targeted RNAs fail to lose m⁶A at the MII stage after *Alkbh5* depletion (Fig. 7i). These data together imply the potential role of IGF2BP2 as a downstream effector in *Alkbh5*-null oocytes.

IGF2BPs, which comprise a family of m⁶A reader proteins, are well known for their functions in the stabilization of m⁶A-modified RNAs in cancer cells, as well as in murine oocytes[38,39]. To gain insight into the mechanisms underlying IGF2BP2-mediated RNA accumulation, we first evaluated the expression patterns of the *Igf2bp* family members *Igf2bp1*, *Igf2bp2* and *Igf2bp3*. The expression levels of *Igf2bp2* and *Igf2bp3* remained unchanged upon *Alkbh5* knockout, except for *Igf2bp1*, which was almost undetectable throughout meiotic maturation (Fig. 8j).

Next, we inhibited *Igf2bp2* expression in oocytes by RNA interference (RNAi). RT-qPCR confirmed that RNAi led to an efficient knockdown (~50%) of *Igf2bp2* mRNA levels (Fig. 8k). As observed above, mock-depleted *Alkbh5*<sup>−/−</sup> oocytes exhibited delayed GVBD onset and strikingly reduced PBE rates at 6 h and 15 h following release from milrinone (Fig. 8l). However, we found that *Igf2bp2* depletion significantly, although incompletely, rescued these defects in *Alkbh5*-null oocytes. It is unlikely that *Igf2bp2* knockdown alone facilitates GVBD and PB1 extrusion, as mock-depleted and *Igf2bp2*-depleted WT oocytes underwent GVBD and completed MI at comparable rates (Fig. 8l), consistent with the phenotypes detected in *Igf2bp2*<sup>−/−</sup> oocytes[39]. *Igf2bp2* depletion also appeared to partially rescue the oocyte quality defects induced by *Alkbh5* loss, as MII oocytes subjected to si*Igf2bp2* microinjection harbored bipolar spindles, although they still exhibited mild chromosome misalignment (Fig. 8m). Notably, the rescue effect was not limited to meiotic cycle progression. As reported previously, a cohort of genes, including *Rpl39*, *Atp5j2* and *Esrrb* (putative IGF2BP2 targets), were resistant to maternal transcript degradation and thus were markedly upregulated in MII oocytes upon *Alkbh5* knockout (Fig. 5d). Conversely, when si*Igf2bp2* was microinjected, we found a significant reduction in the abundance of these transcripts in MII *Alkbh5*<sup>−/−</sup> oocytes compared with that in the mock-depleted *Alkbh5*<sup>−/−</sup> oocytes, which indicated that delayed RNA degradation was attenuated by *Igf2bp2* deletion (Fig. 8n–p). Taken together, these findings showed that global reduction of IGF2BP2 partially overcame the need for timely maternal degradation and reversed the cell cycle defects in *Alkbh5*<sup>−/−</sup> oocytes.

## Discussion

In recent years, the functional roles of m⁶A writers and readers in regulating oogenesis have been documented[12–14,16–18,40]. However, the biological functions of m⁶A erasers in oocyte maturation remain elusive. Here, we demonstrated the essential role of the m⁶A eraser-associated protein ALKBH5 in oocyte meiotic maturation. Using the *Alkbh5* knockout mouse model, we found that *Alkbh5* null mice were infertile and exhibited severe meiotic maturation defects. Importantly, our study captured the abnormal maintenance of m⁶A enrichment in MII oocytes upon *Alkbh5* knockout. This aberrant epigenetic state where m⁶A exclusively persists in a subset of translation-related transcripts results in the RNA accumulation observed in KO oocytes. Mechanistically, the m⁶A reader IGF2BP2 stabilizes a subset of transcripts with persistent m⁶A in KO oocytes (Fig. 9). These results illustrate that ALKBH5 plays a critical role in female reproduction and oocyte maturation.

A previous *Alkbh5*<sup>−/−</sup> mouse line was established by LoxP/Cre-mediated excision of *Alkbh5*[20]. However, in our present study, a new *Alkbh5* knockout mouse strain was generated by ablating *Alkbh5* exon 1 using the CRISPR-Cas9 system. We found that *Alkbh5*<sup>−/−</sup> females were sterile after more than 150 days of continuous breeding with three pairs of matings. Complete infertility was also supported by the total absence of 4-cell embryos derived from KO oocytes at the corresponding time. We did notice that Zheng et al. showed a remarkably low but detectable success rate of breeding (1.46 pups per mating for a total of 26 matings) when *Alkbh5*<sup>−/−</sup> females were crossed with heterozygous males[20]. We believe that this could be justified by the difference in generation methods, female numbers for mating and breeding duration.

The transition from the MI to AI stage in oocytes is driven by activated APC<sup>Cdc20</sup> and thus results in chromosome separation and consequent PB1 extrusion[41]. We found that *Alkbh5* deficiency inactivates APC<sup>Cdc20</sup>, which is partly due to prolonged SAC activation, as shown by the persistent kinetochore localization of BubR1 at the AI stage. Of note, SAC is known as a surveillance system that coordinates kinetochore–microtubule attachment and anaphase onset[42]. Its inhibitory signal is only abrogated after all the chromosomes have achieved the correct attachment to the opposite spindle poles. Focusing on master regulators essential for spindle and chromosome events, we observed RNA accumulation of certain genes in KO MII oocytes. The key genes for the correction of erroneous kinetochore–microtubule attachment in mouse oocytes were upregulated, including *Zwint* and *Cdk7*, which are required for SAC function[43,44]. Genes regulating chromosome kinetics also accumulated, such as *Birc5*, which is required for chromosome segregation[45], and *Septin2*, which is required for chromosome congression[46]. Interestingly, *Birc5* (which encodes the Survivin protein), which has been reported as a critical chromosomal passenger complex (CPC) subunit, showed apparent m⁶A modification in GV oocytes and was proven to be subjected to ALKBH5-mediated stabilization in our follow-up experiments. Moreover, elevated RNA levels of *Ska3*, a member of the spindle and kinetochore-associated complex, were also observed

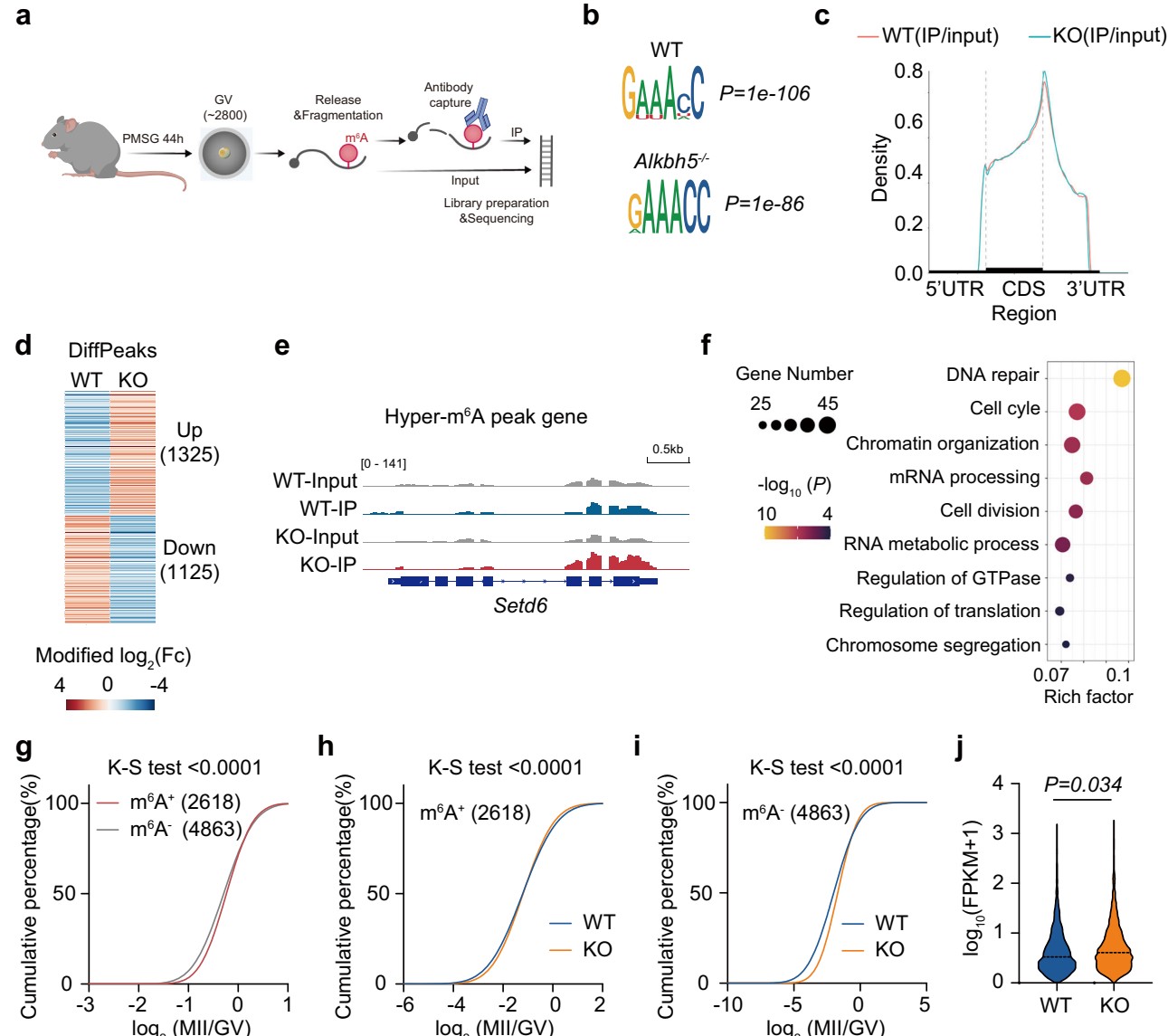

**Fig. 6 | The m⁶A landscape of mouse oocyte upon *Alkbh5* depletion. a** Schematic illustration showing the pipeline of m⁶A meRIP-seq as detailed in "Methods". In brief, about 2800 GV oocytes are collected from WT and *Alkbh5⁻/⁻* mice after PMSG injection, and are prepared for m⁶A methylome analysis. **b** Examples of m⁶A-specific motifs detected from WT and *Alkbh5⁻/⁻* samples. The consensus motif of m⁶A "RRACH" (R = G or A; H = A, C, or U) observed in the m⁶A-seq are illustrated along with *P* values analyzed by HOMER. **c** The m⁶A-peak distribution of *Alkbh5⁻/⁻* samples compared with the controls. **d** Heatmap showing differential hyper- and hypo- m⁶A peaks (hyper, DiffModLog₂Fc > 1 with *P* < 0.01; hypo, DiffModLog₂Fc < −1 with *P* < 0.01) after *Alkbh5* deletion. Each m⁶A peak is color-coded from blue to red according to their DiffModLog₂Fc values. *P* values are calculated based on the Poisson distribution. **e** Representative IGV diagram of m⁶A distributions of the hypermethylated gene *Setd6*. **f** GO analysis of genes containing differentially changed m⁶A peaks following *Alkbh5* knockout. Functions of interest are annotated and numbers of genes enriched are also labeled. The enrichments and *P* values are calculated using hypergeometric test of functional annotation in DAVID website with default parameters. **g** Cumulative fraction showing the difference in degradation rates (log₂[fold change (MII/GV)]) between the m⁶A-modified and unmodified RNAs in WT oocytes. *P* values, one-sided Kolmogorov-Smirnov test. **h** Cumulative fraction showing the degradation rates (log₂[fold change (MII/GV)]) of m⁶A-modified RNAs in WT and *Alkbh5⁻/⁻* oocytes, respectively. *P* values, one-sided Kolmogorov–Smirnov test. **i** Cumulative fraction showing the degradation rates (log₂[fold change (MII/GV)]) of unmodified RNAs in WT and *Alkbh5⁻/⁻* oocytes, respectively. *P* values, one-sided Kolmogorov-Smirnov test. **j** Violin plot showing the expressional levels of m⁶A-modified transcripts in WT and *Alkbh5⁻/⁻* oocytes at MII stage. The data are presented as violin with central line denoting mean value. *P* values, two-tailed Mann-Whitney test. Source data are provided as a Source Data file.

in KO MII oocytes, and its role in controlling spindle stability has been well studied[47]. In addition, among these accumulated transcripts, CDC25B is of particular interest, as it is well established as a key phosphatase that mediates CDK1 dephosphorylation and thus governs cell cycle progression. Its accumulation could contribute to spindle and chromosome abnormalities in mouse oocytes[48]. Based on our RNA-seq dataset, the *Cdc25b* RNA levels in KO oocytes were comparable to those in WT oocytes until GVBD. However, *Cdc25b* was observed with a markedly higher expression level in KO MII oocytes.

Thus, our study highlights the potential explanations for persistent SAC activation and resultant MI arrest, in which accumulation of these transcripts might confer spindle and chromosomal defects to the KO oocytes.

Recent knowledge of the m⁶A landscapes of mouse oocytes and embryos led us to reconsider the role of m⁶A in the regulation of RNA clearance machinery. Before fertilization, m⁶A modification could slow the degradation rate of maternal transcripts and stabilize maternal mRNA in mouse oocytes[16,17]. In line with previous findings, we found a

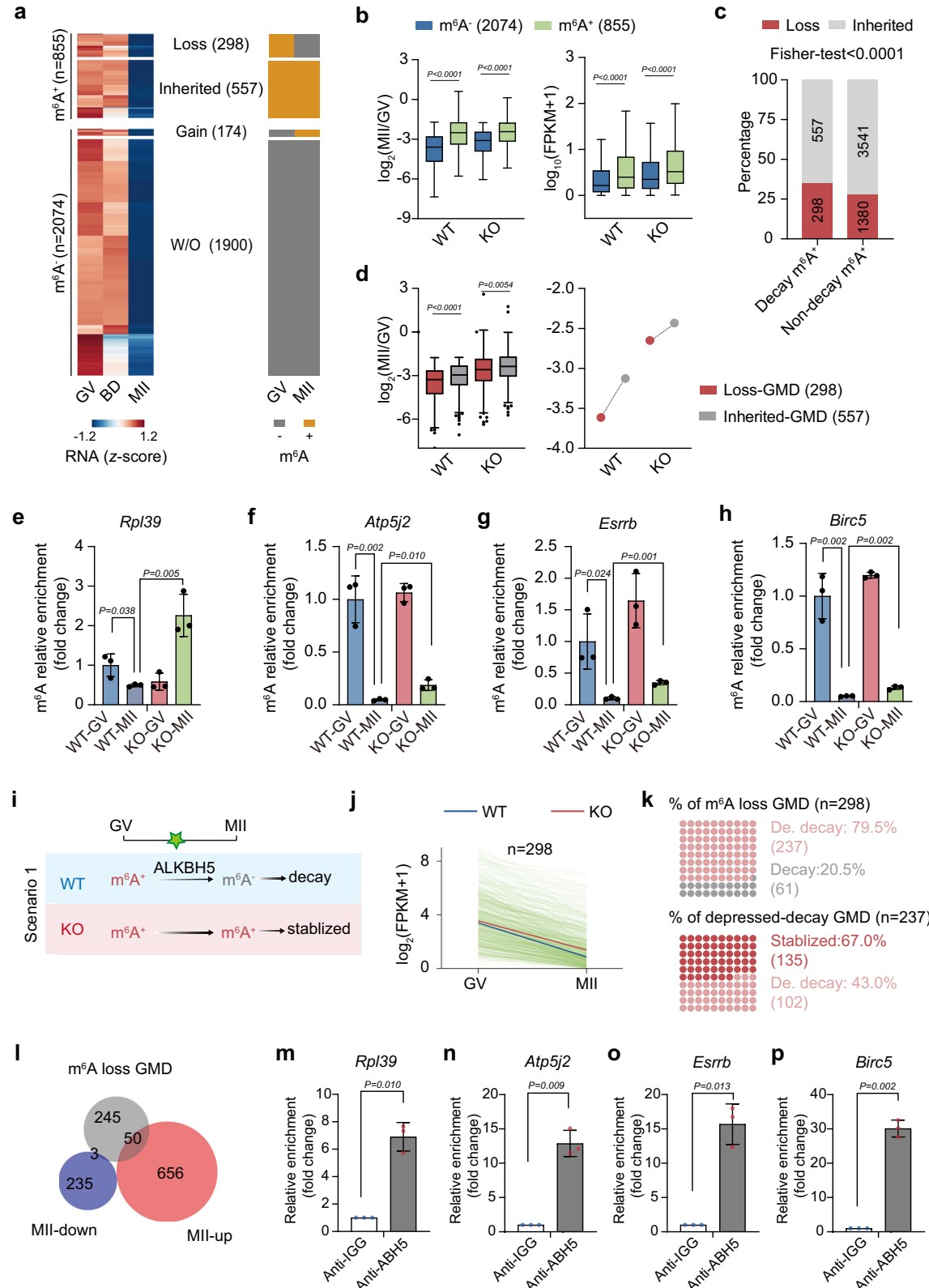

substantial delay in the maternal decay of m6A-marked RNAs and inherited-GMD transcripts compared with their unmarked and loss-GMD counterparts, respectively, during the GV-to-MII transition. We speculate that loss-GMD transcripts may be the substrates of ALKBH5. In support of this notion, we observed a degradation dysfunction of loss-GMD transcripts in *Alkbh5*[−/−] oocytes and validated the binding of ALKBH5 to representative transcripts. However, ALKBH5 substrates

were not indeed comprehensively identified in our study due to the lack of RNA-RBP interaction analysis. Further investigations are needed to clarify this point.

Interestingly, whereas m6A plays a stabilizing role before fertilization, after zygotic genome activation (ZGA), transcripts with m6A are degraded more quickly than those without m6A[16,17]. The opposite effects of m6A modification on maternal transcripts in the two stages

**Fig. 7 | ALKBH5-mediated demethylation ensures the timely decay of maternal RNAs. a** Heatmaps showing the expression levels of GV-MII decayed transcripts (abbreviated as "GMD") in WT oocytes during GV-MII transition. The rightmost columns indicate the m⁶A marking status of GMD. **b** Boxplots showing the expression fold changes (left) and RNA abundance (MII stage, right) of m⁶A-modified and unmodified transcripts in oocytes. $n = 2074$ (m⁶A⁻), 855 (m⁶A⁺) transcripts. $P = 6.94e-4, 0.000006, 3.03e-9$ and $0.000037$ (from left to right). **c** Proportions and numbers of m⁶A-loss and inherited transcripts in decayed and non-decayed m⁶A-modified transcripts. $P$ values, two-tailed Fisher's exact test. $P = 0.000074$. **d** Boxplot (left) and dot plot (right) showing the expression fold changes of loss and inherited GMD transcripts in WT and $Alkbh5^{-/-}$ oocytes. The dot represents the mean value of expression fold changes. WT, $P = 1.62e-8$. **e–h** RNA immunoprecipitation results showing the relative m⁶A enrichments among the indicated genes in GV and MII oocytes from WT and $Alkbh5^{-/-}$ mice. The data validated by qPCR are present as mean ± SD and are normalized to WT GV samples.

$P$ value, two-tailed Student's t test. $n = 3$ biological replicates. **i** A schematic model of ALKBH5 function during GV-MII transition. **j** Degradation patterns of loss-GMD transcripts during GV-MII transition. Each green line represents the expression level of an individual gene. The blue and red line represent the median expression levels in WT and $Alkbh5^{-/-}$ oocyte, respectively. **k** Dot plots showing percentages of repressed decayed genes in loss-GMD ($n = 298$) and stabilized genes in affected transcripts ($n = 237$). De. Depressed. **l** Venn diagram showing the overlapping of loss-GMD with DEGs in $Alkbh5^{-/-}$ MII oocytes. **m–p** RIP-qPCR analysis showing the fold changes of ALKBH5 binding ability with the indicated RNAs in oocytes. The bar is mean ± SD resulting from three independent experiments. $P$ value, two-tailed Student's t test. Data in **b** and **d** are presented as box and whisker plots. The central line denotes median value, while the bounds of box mark the 25th to 75th percentiles and whiskers refer to minimum and max values by Tukey. $P$ values in **b** and **d** are calculated by two-tailed unpaired Wilcoxon test. Source data are provided as a Source Data file.

illustrate the different molecular mechanisms mediated by m⁶A in oocyte and early embryo development. Given that $Alkbh5^{-/-}$ oocyte-derived embryos arrest at the 2-cell stage, we speculated that ALKBH5 might also function in maternal RNA turnover during the MZT. Further studies are warranted to understand how ALKBH5 regulates maternal RNAs after fertilization.

The functional consequences of m⁶A are mediated by m⁶A reader proteins. Although there are various types of readers, certain proteins have been shown to be associated with RNA stability in previous studies, including YTHDF2, YTHDC1 and IGF2BPs[10,13,17,33,34]. Thus, we reasoned that one of these readers is the likely mediator of the observed RNA decay defect in $Alkbh5$ KO oocytes. Notably, we found that m⁶A-modified regions of loss-GMD were significantly accessible to IGF2BP2 and 3, especially IGF2BP2. Consistent with this, we demonstrated using RIP-qPCR that IGF2BP2 could bind to the selected transcripts in mouse oocytes, while enrichment of other readers at these transcripts was markedly low. We further extended the IGF2BP2 enrichment to MII oocytes and found that IGF2BP2 enrichment at these transcripts was markedly disrupted in WT oocytes but not in KO oocytes. These results suggest that IGF2BP2 could be the regulator in the degradation of some certain maternal RNAs. Further RNA-RBP interaction study for $Alkbh5^{-/-}$ oocytes are needed to identify the binding targets of IGF2BP2 for maternal RNA stabilization. Additionally, the following observation that knockdown of endogenous $Igf2bp2$ expression in $Alkbh5^{-/-}$ oocytes rescued meiotic maturation defects further supported our hypothesis. However, we could not rule out the possibility that a small number of transcripts may be bound by other readers, since this process is complicated and still unclear. Future studies should explore the possibility that other readers may participate in maternal mRNA accumulation caused by $Alkbh5$ deficiency.

## Methods
A detailed description of materials and bioinformatics analysis methods are available in the Supplementary Data 7–9.

### Animals
The experimental mice were all of *C57BL/6* genetic background, $Alkbh5^{+/-}$ mice were designed and purchased from Gem Pharmatech. Animals were maintained under appropriately controlled environment (12 h light/12 h dark cycle with temperature of 18–23 °C) and humidity (50–70%) and with easy access to food and water. All animal experiments were approved by local animal ethnical committee and were strictly conducted in compliance with the Animal Care and Use Committee of Zhejiang University (File no. ZJU20220520). Fertility analysis was performed for more than 150 days of continuous breeding using three pairs of 2-month-old $Alkbh5^{-/-}$ female and wild-type male.

### Ovarian histology
Ovaries were dissected from WT and $Alkbh5^{-/-}$ female mice aged 3-week and fixed overnight in 4% PFA. Samples were then embedded in paraffin followed by serial section in thickness of 5 μm. After incubation with ALKBH5-specific antibody and its corresponding secondary antibody, sections were stained with hematoxylin and mounted in neutral balsam for observation.

### Superovulation, oocyte collection and fertilization
For collection of GV oocytes, 21-day females were injected with 10 I.U. pregnant mare serum gonadotropin (PMSG) and then sacrificed 48 h later. To release cumulus-oocyte complex (COC), hormone-stimulated ovaries were dissected in M2 medium (Sigma, M7167) supplemented with penicillin–streptomycin (Gibco, 15140-122) and 2.5 μM milrinone (Sigma). Repeated mechanical aspiration allowed for isolation of denuded oocyte from surrounding cumulus cells. Only full-grown GV oocytes with good morphology were chosen and then cultured.

In order to harvest in vivo MII oocytes, superovulation was performed on female mice by injection of 10 I.U. PMSG followed with 10 I.U. human chorionic gonadotropin (hCG) injection 44 h later. 16 h after hCG treatment, COCs were obtained through puncturing oviduct ampullae with 30-gauge needles. Somatic cells were removed by brief digestion with hyaluronidase.

For fertilization, the super-ovulated female mice of indicated genotypes were mated with 10–12-week-old WT males. Successful mating was confirmed by the presence of vaginal plugs.

### Oocyte culture and drug treatment
To inhibit meiotic resumption, collected oocytes were maintained in M16 medium containing 2.5 μM milrinone and penicillin-streptomycin under mineral oil (Sigma, M8410) in a dark and humidified atmosphere with 5% $CO_2$ at a constant temperature of 37 °C. After washing through consecutive drops of M16 without milrinone, oocytes were released in inhibitor-free medium for in vitro culture. In some experiments mentioned above, Cyclin-dependent kinase (CDK) inhibitor roscovitine was added 6 h after onset of germinal vesicle breakdown (GVBD) in order to constrain maturation promoting factor (MPF) activity. For overriding spindle activation checkpoint (SAC), oocytes were exposed to 10 μM reversine 2 h after observation of GVBD.

### Zygote collection and embryo culture
Mouse zygotes were harvested from oviducts of super-ovulated female mice that were mated with WT males 28 h after hCG injection. The fertilization rate was assessed by the presence of two pronuclei. After selection, zygotes were cultured in drops of KSOM medium (Millipore, MR-121) under mineral oil at 37 °C in a 5% $CO_2$ atmosphere. 2-cell and 4-cell embryos were obtained at 40 and 54 h post hCG injection, respectively.

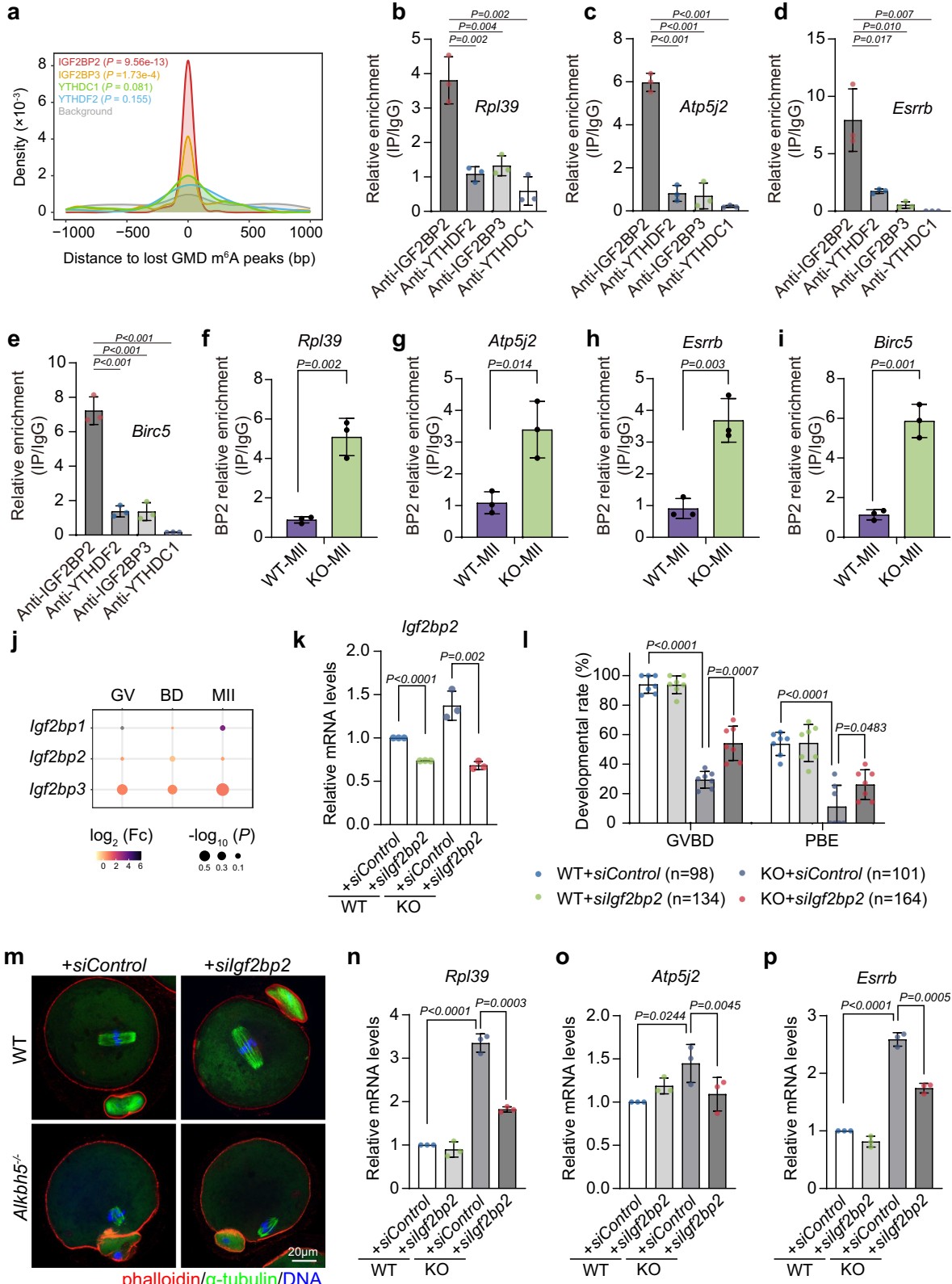

## Cell culture

For primary mouse granulosa cell (mGC) collection, female mice of respective genotype aged 3-week were primed with PMSG and sacrificed 44 h after injection. mGCs were released through manually puncturing ovarian follicles. Cells were cultured in Dulbecco's Modified Eagle's Medium/F-12 (Sigma, D2906) containing penicillin-streptomycin supplemented with 10% fetal bovine serum (Vistech, SE200-ES) and GlutaMAX (Gibco, 35050061) at 37 °C under 5% $CO_2$ with controlled humidity.

## Small interference RNA microinjection

Small interference RNA (siRNA) of murine *Alkbh5* and *Igf2bp2* were purchased from Santa Cruz (sc-141022) and GenePharma respectively. For gene knockdown in cell lines, mGC was transfected with siRNA via

**Fig. 8 | The m⁶A reader IGF2BP2 is involved in the maturation defects caused by *Alkbh5* knockout. a** The density plot showing the distance between binding sites of m⁶A readers (IGF2BP2, IGF2BP3, YTHDF2 and YTHDC1) detected by RIP-seq and eCLIP-seq in mouse and human ESCs and m⁶A peaks of loss-GMD detected by m⁶A-seq in our dataset. **b–e** RIP-qPCR results showing the relative enrichments of the indicated m⁶A readers among the targeted genes in oocytes. The data validated by qPCR are present as mean ± SD and are normalized to IgG samples. For *P* values in **c** *P* = 0.000092, 0.000239 and 0.000020; in **e** *P* = 0.000302, 0.000439 and 0.000107 (from left to right). **f–i** RIP-qPCR results showing the relative enrichments of IGF2BP2 among the targeted genes in WT and *Alkbh5⁻/⁻* MII oocytes. The data are presented as mean ± SD and are normalized to IgG samples. **j** Bubble plot showing the expression fold changes of *Igf2bp1*, *Igf2bp2* and *Igf2bp3* during oocyte maturation upon *Alkbh5* depletion detected by RNA-seq. Genes differential expression analysis was performed by DESeq2 software between two different groups. **k** RT-qPCR results showing the knockdown efficiency of *Igf2bp2* RNA

interference in WT and *Alkbh5⁻/⁻* oocytes respectively. The bar is mean ± SD resulting from three independent experiments. WT, *P* = 0.000034. **l** The rates of GVBD and PB1 emission in WT and *Alkbh5⁻/⁻* oocytes microinjected with control and *Igf2bp2* siRNA. Numbers of oocytes examined are indicated. Data are presented as mean values ± SD. GVBD, *P* = 8.78e−11; PBE, *P* = 0.000068. **m** Representative confocal images of spindle assembly and chromosome alignment in oocytes microinjected with control and *Igf2bp2* siRNA, respectively. Spindle is labeled in green by α-Tubulin, while DNA is marked in blue by DAPI. Phalloidin (red) highlights the margin of oocyte. **n–p** RT-qPCR results showing the relative mRNA levels of the indicated genes in MII oocytes microinjected with control and *Igf2bp2* siRNA. The bar is mean ± SD resulting from three independent experiments. For *P* values in **n** *P* = 0.000043; in **p** *P* = 0.00002. Data in **b-i** represent results from three biologically independent experiments. *P* values in **b-i**, **k-l** and **n-p** were calculated by two-tailed Student's t test. Source data are provided as a Source Data file.

Lipofectamine™ RNAiMAX (Invitrogen, 13778150). Protein analysis was performed 48 h after transfection. For RNA interference in oocytes, approximately 5–10 pl siRNA diluted with RNase-free water (60 µM) was microinjected into GV oocytes as detailed in the followings.

### In vitro mRNA synthesis and microinjection
The *Alkbh5* (NM_172943.4) open reading frame sequence was obtained by PCR-based amplification with appropriate primers (Supplementary Data 7) using Q5® High-Fidelity DNA Polymerase amplify system (NEB, M0491). The original vector pRK5-Flag-eGFP (a gift from Dr. Hengyu Fan) contains a SP6 promoter which allows for in vitro transcription, and multiple cloning sites including XhoI and XbaI. In our case, XhoI and XbaI were selected as cloning sites and obtained *Alkbh5* sequence was fused into the vector by restriction digestion and ligation using restriction endonucleases (NEB, R0146V and R0145V) and T4 DNA ligase (NEB, M0202S). The fidelity of reconstructed vector was confirmed by DNA sequencing. Capped mRNAs encoding fluorescently labeled full-length of *Alkbh5* coding region were in vitro transcribed from reconstructed pRK5-Flag-eGFP (a gift from Dr. Heng-yu Fan) using mMACHINE™ SP6 kit (Invitrogen, AM1340) according to manufacturers' guidelines. After polyadenylation by Poly(A) Tailing kit (Invitrogen, AM1350), the newly synthesized cRNA were purified with MEGAclear™ Kit (Invitrogen, AM1908) and eluted in RNase-free water.

For microinjection, approximately 5–10 pl synthesized cRNA (800 ng/µl) or diluted siRNA (60 µM) were injected to the cytoplasm of full-grown oocytes with intact GV with the appliance of Eppendorf Transferman NK2 micro-manipulators. Dead oocytes or with bad morphology were excluded soon after injection to exclude the effects of mechanical injury. Injected oocytes were then cultured in M16 medium with milrinone for 6 h or 12 h to allow for protein translation or targeted RNA knockdown, respectively. Following incubation, oocytes were released into milrinone-free medium for in vitro maturation. The ratio of GVBD and PBE were calculated 6 h and 16 h later.

### RT-qPCR
5–10 oocytes collected were lysed in cell lysis buffer containing 0.2% Triton X-100 and RNase inhibitor after removal of zonal pellucida exposed to acidic M2 medium. The lysate was then reverse transcribed using SuperScript III First-Strand System (Invitrogen, 18080051) following manufacturers' instructions. Quantitative PCR was conducted with One Step TB Green® PrimeScript™ RT-PCR Kit (Takara, RR066A) on ABI QuantStudio5 Real-Time PCR system (Applied Biosystems). Murine *Atcb* was applied as an endogenous control since it remained stable during GV-MII transition detected by RNA-seq. Details of oligonucleotide primers used in this project were listed in Supplementary Data 7. Data were obtained from at least three replicated biological experiments per genotype with three technical repeats each time and were expressed as enrichment of $2^{-\Delta\Delta Ct}$[49].

### Western blot
Pooled oocytes at indicated time points were sequentially washed in 0.1% PVP in PBS for three times followed by lysis in 2×lysis buffer (CST, 9803) supplemented with protease inhibitor cocktails. The protein extracts were then boiled at 95 °C for 5 min in the mixture of loading buffer and β-mercaptoethanol at 5:1 ratio for denaturation. Targeted proteins were separated on 8% v/v polyacrylamide gels and subsequently transferred to PVDF membrane (Roche, 43487300) via wet transfer for 2 h (200 mA). Prior to incubation with primary antibody overnight, membranes were blocked with 5% non-fat milk dissolved in TBS-Tween 20 (0.05%) for 1 h at room temperature. After three washes in TBST for a total of 1 h, membranes were incubated with HRP-conjugated secondary antibodies for 1 h at room temperature. Detection of protein signals was performed using Pierce™ ECL Western Blotting Substrate (Thermo, 32106) and bands intensity was measured by Image J software. The primary antibodies and dilutions used are described in Supplementary Data 8. The uncropped and unprocessed scans of blots are provided in the Source Data file.

### PAT assay
Analysis of poly(A) tail lengths by PCR was conducted according to the published protocol[50]. In brief, total RNA was isolated from about 200 oocytes at GV, MI, MII stages using RNeasy Mini Kit (QIAGEN, 74106) and resolved in RNase-free water with expectation of concentration on 5–10 ng/µl. Appropriate amount of RNA was mixed with 20 ng oligo(dT)₂₀ and then heated at 65 °C for 10 min for denaturing. Specifically designed oligo(dT)-anchor with high GC-enrichment (5′-GCG AGC TCC GCG GCC GCG T12) was annealed and ligated to 3′ poly(A) overhangs via high concentrations of T4 DNA Ligase (NEB, M0202T) followed by in vitro reversed-transcriptions by SuperScript III First-strand Synthesis Super Mix. The newly synthesized cDNA with oligo(dT)-anchor tagged at 5′ end was then subjected to PCR using P2 of targeted genes as primers. The specific sequence of primers designed were shown in Supplementary Data 7.

### Detection of nascent RNA and protein synthesis in oocytes
The global transcription activity was observed utilizing Click-iT® RNA Imaging Kits (Invitrogen, C10330). Oocytes with germinal vesicle were harvested from 3-week-old female mice and incubated with 5-ethynyl uridine (5-EU) in M16 medium containing milrinone for 1 h, enabling the efficient incorporation of labeled nucleoside into nascent RNA. For detection of protein changes, L-homopropargylglycine (HPG) was added into the culture medium for 30 min incubation with MII oocytes. The signals of newly synthesized proteins were visualized temporally and spatially utilizing Click-iT® HPG Alexa Fluor® Protein Synthesis Assay Kits (Invitrogen, C10428).

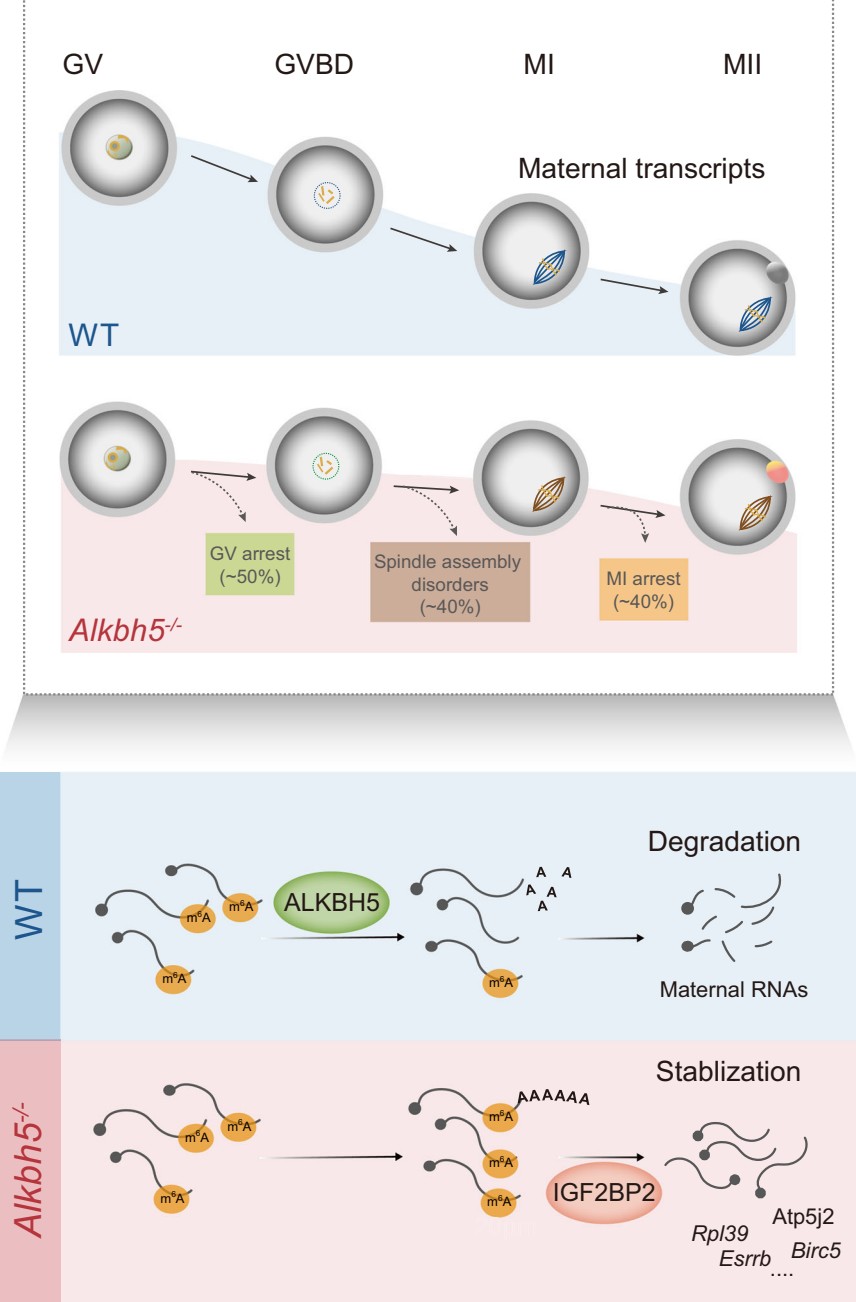

**Fig. 9 | ALKBH5 manipulates maternal RNA clearance in GV-MII transition by regulating m⁶A demethylation.** Illustrations summarizing the functions of ALKBH5 in oocyte maturation and the molecular mechanism regarding how ALKBH5 regulates maternal RNA decay in oocytes.

## Chromosome spreads

Zonal pellucida of oocytes with corresponding stages was removed by exposure to acidic M2 medium. After recovery for 10 min at 37 °C, zonal pellucida-free oocytes were then transferred to a clean glass slide dipped with one drop of spreading solution (1% paraformaldehyde, 0.15% Triton X-100, 3 mM dithiothreitol). Slides were allowed to dry slowly in a humid chamber overnight.

## Immunofluorescence staining

For immunolabeling of spindles and MTOC, oocytes at indicated stages were fixed in freshly made 2% PFA for 30 min and then permeabilized by 0.25% Triton X-100 for 15 min. After block in 0.3% BSA for 30 min, samples were incubated with primary antibodies in a humidified chamber overnight at 4 °C. Primary antibodies applied here were: α-Tubulin-FITC (Sigma, F2168), TPX2 (Abcam, ab252944), Pericentrin (BD

transduction laboratories, 611814). Oocytes were then transferred through sequential three drops of PBST to wash out of primary antibodies and exposed to a 30 min incubation of corresponding secondary antibodies at room temperature. Samples were counterstained by DAPI for DNA detection and mounted with Slow Fade Diamond Antifade Mountants (Invitrogen, S36967). For staining on chromosome spreads, slides were blocked with 0.3% BSA for 30 min and incubated with primary antibodies against Crest, BubR1 at 4 °C overnight, and then washed, stained with secondary antibodies as mentioned above. Images were acquired on Lecia SP8 laser-scanning confocal microscope imaging system from three repeated experiments with equal setting per repeat. For analysis of BubR1, three random BubR1 signals were calculated using ImageJ in each oocyte and were normalized to Crest intensity of the same kinetochore. The primary antibodies and dilutions used are described in Supplementary Data 8.

## RNA immunoprecipitation

The RNA immunoprecipitation assay (RIP) was performed according to the published protocol[51]. Briefly, 200 oocytes at indicated stages were collected from WT and *Alkbh5*−/− females and dissolved in ice cold homogenization buffer (150 mM KCl, 2 mM EDTA, 0.5% NP-40, 0.5 mM DTT, 50 mM HEPES (pH 7.5) and protease inhibitor cocktail prior to centrifugation at 12000 rpm for 5 min. Supernatant was subsequently pre-cleared by incubating with washed Protein A/G PLUS-Agarose (Santa Cruz, sc-2003) for 30 min and 10% of cell lysate was aliquoted for input. The remaining samples were incubated with 5 μg antibodies against ALKBH5, IGF2BP2, IGF2BP3, YTHDF2, YTHDC1 or its corresponding IgG as negative control for 4 h at 4 °C on a rotating device. After pre-incubation, pre-washed protein A/G beads were added to the mixture followed by an overnight incubation with rotation at 4 °C. On the following day, bead pellets were washed for four times with washing buffer. Immunoprecipitated RNA was purified utilizing RNeasy® Mini Kit (Qiagen, 74106). In RIP assay for m6A, total RNA was extracted from 200 oocytes at indicated stages of each genotype, and then directly incubated with m6A-antibody (Synaptic Systems, 202-003) on a rotator for 4 h at 4 °C. Immunoprecipitated RNA was purified using RNeasy® Mini Kit. Both input and IP samples were reversely transcribed and real-time qPCR were conducted as detailed previously. Data were calculated as relative enrichment, with IgG samples set to 1. For mGC RIP assay, cells were obtained from ovaries of PMSG-primed females and dissolved in ice cold homogenization buffer. The following RIP and analysis process were the same as in the oocyte RIP assay protocol mentioned above. The primary antibodies and dilutions used are described in Supplementary Data 8.

## RNA-Seq sample collection and library preparation

3-week-old female WT and *Alkbh5*−/− mice were treated with 10 IU PMSG by intraperitoneal injection and sacrificed 44 h later. GV stage oocytes with good morphology were selected and cultured in vitro as mentioned above. 10 oocytes were obtained at 0 h, 4 h and 16 h as GV, GVBD, MII samples and their zonal pellucida were removed by a brief incubation with acidic M2 medium. After rinse in M2 medium and 0.1% BSA in PBS, samples were transferred to PCR tubes and snap frozen, stored in −80 °C until sequencing. Total RNA was released in cell lysis buffer supplemented with RNase inhibitor and 0.2 μl ERCC RNA Spike-In Mix (Invitrogen, 1:1000 diluted) were added to each tube prior to retro transcription. Polyadenylated mRNA were reversely transcribed for first-strand cDNA synthesis using SMARTScribe™ Reverse Transcriptase. The following poly(A) tailing and PCR amplification were performed according to instructions[52]. Briefly, after the first-strand reaction, the cDNA was pre-amplified by 12 cycles PCR reaction to obtain the full-length cDNA products through IS PCR primer-mediated semi-suppressive PCR. Then, cDNA was fragmented by dsDNA Fragmentase (NEB, M0348S) incubating at 37 °C for 30 min and size-selected by magnetic beads with expectation for 150-300 bp fragments. Libraries were constructed with sheared cDNA using the SMART-Seq v4 Ultra Low Input RNA Kit (Takara, Japan) and then sequenced on Illumina Novaseq™ 6000 system (PE150).

## Bioinformatics analysis

Quality control was performed on raw sequencing data. Poor quality reads (More than 5% of N base; more than 20% of low-quality base (quality score Q < 10)) together with adapter sequence and A/G-enriched reads were trimmed using Cutadapt. Valid data were then aligned to mouse reference genome (GRCm38/mm10) plus ERCC Spike-In. Fasta by Hisat2 with default parameters and only uniquely mapped sequence sorted by SAMtools were subjected to downstream analysis (Supplementary Data 4). Index of each group were recreated utilizing Stringtie. The resulting sequence reads were quantified as Fragments Per Kilobase Million (FPKM) and normalized to the read counts of ERCC. Only transcripts with FPKM > 1 in at least one group were kept

for the following calculation. Differential expression analysis was accomplished using DESeq2 for comparison and BH for correction of *p* value. Gene with |log2fc| ≥ 1 and q (adjusted *p* value) < 0.05 was considered as significantly changed. Note that ERCC normalization were applied to all differential analysis in RNA-seq unless specified. Functional annotation was complemented using DAIVD website and Metascape. Heatmaps, volcano plots and scattered diagrams were constructed with R (v.3.1.1) package gplots.

## Library construction of m6A-MeRIP-Seq

Approximately 1 μg RNA were isolated from about 2800 GV oocytes of hormone-stimulated females using TRIzol following manufacturers' procedures. NanoDrop ND-1000 and Bioanalyzer 2100 were utilized to test quality and integrity of obtained RNA with expectation of RIN > 7, OD260/280 > 1.8. After two rounds of purification, mRNA with poly(A) tails were captured by Dynabeads Oligo (dT) (Thermo Fisher, 25-61005) followed by fragmentation at 86 °C for 7 min using NEBNext® Magnesium RNA Fragmentation Module. The resulting cleaved RNA were then incubated with Dynabeads (Thermo Fisher) conjugated to m6A-specific antibody in IP buffer (50 mM Tris-HCl, 750 mM NaCl and 0.5% Igepal CA-630) overnight with rotation. Immunoprecipitates were reverse transcribed to cDNA with SuperScript™ II Reverse Transcriptase (Invitrogen, 1896649) after extensive washing. Ligated with dual-index adapters, fragments with size of 180–220 bp were sorted and purified by magnetic beads. Sequence libraries were established through PCR amplification under following conditions: pre-denaturation at 95 °C for 3 min, denaturation at 98 °C for 15 s per cycle (8 cycles), annealing at 60 °C for 15 s and extension at 72 °C for 30 s with final extension for 5 min. Sequencing of m6A was finally performed on illumina Novaseq™ 6000 (LC Bio Technology CO., Ltd. Hangzhou, China) under double-end, 150-base pair conditions according to the standard protocols.

## Data analysis

Raw data were processed by fastp software and reads with adapter contamination, low quality, or repeated sequence were filtered out. HISAT2 was then utilized for mapping clean reads to mouse reference genome *Mus musculus* (Version: v101) (Supplementary Data 10). The resulting bam files were subjected to peaking calling and visualization with R package exomePeak2 (https://bioconductor.org/packages/release/bioc/html/exomePeak2.html) followed by annotation of m6A peaks (IP/input>2, P < 0.05) using ANNOVAR. De novo motif enrichment was accomplished with appliance of MEME2 (http://meme-suite.org) and HOMER (http://homer.ucsd.edu/homer/motif). Differential m6A peaks were defined as peaks meeting the criterion of |log2 FC| ≥ 1 and *p* value < 0.05, with their corresponding gene expression >1 in WT samples. For comprehensive analysis of m6A-seq and RNA-seq, transcripts with WT input FPKM > 1 containing at least one exonic m6A peak were considered as m6A-modified genes. An m6A-modified transcript was defined as ALKBH5-dependent (A5+ m6A+) if m6A peaks were detected in *Alkbh5*−/− samples but absent from WT samples; otherwise, it was named A5+ m6A+ transcript. IGV software was used to demonstrate the m6A peak distribution among the indicated transcripts with bw files. Analysis of cumulative fractions were performed using GraphPad Prism8 and their statistical significance were calculated by Kolmogorov-Smirnov test.

## m6A reader analysis

The binding peaks of IGF2BP2/3 identified in hESCs were extracted from eCLIP-seq data (GSE78509)[35]. The genome coordinates between human and mouse were conversed by UCSC liftOver. YTHDF2 binding peaks identified by eCLIP-seq in mESCs were downloaded from GEO database (GSE151788)[36]. YTHDC1 binding peaks identified by RIP-seq in mESCs were obtained from GEO database (GSE157268)[37]. Distance between m6A readers' targets and identified m6A peaks in our study was calculated utilizing the "closet" of BEDTools. Background was

obtained by randomly permute the genomic locations of m⁶A reader binding peaks using shuffleBed of BEDTools.

## Comparative analysis of m⁶A-seq datasets

The m⁶A profiles of GV oocytes from picoMeRIP-seq dataset were downloaded from Gene Expression Omnibus (GEO; dataset accession number 192440). The m⁶A sequencing data of single GV oocyte detected by scm⁶A-seq were downloaded from Genome Sequence Archive of National Genomics Data Center (accession code CRA006425). The data processing steps were the same as previously described and peaks with IP/input ratios greater than 2 ($P < 0.05$) were defined as m⁶A peaks. For defining m⁶A-modified genes in different datasets, genes with an input FPKM > 1 containing at least one exonic m⁶A peak were defined as m⁶A-modified genes; otherwise, they were defined as unmodified genes. For definition of integrated m⁶A-modified genes, genes meeting the criteria of both identified m⁶A-modified genes by our m⁶A-seq and overlap with at least one public m⁶A datasets mentioned above were defined as integrated m⁶A-modified genes.

## Gene group categorization

GMD transcripts were defined as transcripts with fold change values (GV/MII ≥ 2, $q < 0.05$). The m⁶A dynamics of GMD transcripts from the GV to MII stage were calculated from public dataset by picoMeRIP-seq (GSE192440) and GMD transcripts were further classified accordingly. Specifically, GMD transcripts were classified by the following formulas: Loss: m⁶A⁺ at GV stage and m⁶A⁻ at MII stage. Inherited: m⁶A⁺ at GV stage and m⁶A⁺ at MII stage. Gain: m⁶A⁻ at GV stage and m⁶A⁺ at MII stage. W/O: m⁶A⁻ at GV stage and m⁶A⁻ at MII stage.

## Statistics and Reproducibility

All statistical analysis involved in this project were conducted using GraphPad Prism 8 unless specified. Calculation methods applied were detailed in respective figure legends. In sum, for comparison between two groups, two-tailed Student's t test or unpaired Wilcoxon test was used to determine $P$ value for appropriate purpose. To estimate statistical significance of cumulative fractions, Kolmogorov–Smirnov test was utilized unless otherwise stated. Genes differential expression analysis was performed by DESeq2 software between two different groups. For GO analysis, enrichments and $P$ values were calculated using hypergeometric test of functional annotation in DAVID website with default parameters. Besides, Image J software was used for signal measurement and data collection. Not significant: $P \geq 0.05$.

The numbers of biologically independent samples and experiments examined in this manuscript can be found in respective figure legends and figures. Each experiment involved in this project were repeated independently for at least three times unless otherwise stated. Western blot of ALKBH5 protein in WT and KO oocytes was repeated independently for three times with similar results. Immunostaining for spindle and chromosome in MI oocytes was performed for three times. Chromosome spreads of MI and MII oocytes, and fluorescence staining of oocytes after roscovitine treatment were repeated independently for three times with similar results. Immunofluorescence of spindle and chromosome morphologies after reversine treatment was performed for three times. PAT analysis was performed once owing to limited availability of KO MII samples. Immunofluorescence of spindle and chromosome after microinjection was performed with three biologically independent replicates.

## Reporting summary

Further information on research design is available in the Nature Portfolio Reporting Summary linked to this article.

## Data availability

The raw data MeRIP-seq and RNA-seq data have been deposited in NCBI's Gene Expression Omnibus (GEO) and are accessible through GEO Series accession number GSE229771 and GSE229774, respectively. All the MeRIP-seq and RNA-seq data generated in this study are summarized in Supplementary Data 1–6 and 10. Source data are provided with this paper.

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

## Acknowledgements

This work is supported by the Key Research and Development Program of Zhejiang Province (2021C03100) to and Natural Science Foundation of China (82071604) Y.Z, Natural Science Foundation of China (81901443) and Natural Science Foundation of Zhejiang province, (LQ19H040013) to L.B., Natural Science Foundation of China (82101759) and Natural Science Foundation of Zhejiang province (LQ22H040006) to M.T. We thank Chao Bi and Xiaoli Hong from the Core Facilities, Zhejiang University School of Medicine, for their technical support. We also thank the staffs from LC-Bio Technology for their sequencing services.

## Author contributions

L.B. and Y.Z. conceived the project and designed the experiments. L.B., Y.X. and M.T. performed and analyzed the bulk of the experiments. S.L., M.Z., Qin.C. and Qic.C. perform the animal feeding and experiments. Sh.W. and Y.Sa. help to collect and culture the primary mouse granulosa cells. L.B., Y.X. and Q.L. performed and analyzed bioinformatics data. Si.W., Z.L. and Y.So. help to perform the microinjection experiments. L.M., X.H. and G.F. help to perform the experiments. L.C. and Y.Y. help to perform the oocyte in vitro culture. L.B., Y.X. and M.T. wrote the manuscript and Y.Z. revised it.

## Competing interests

The authors declare no competing interests.
