## [Peer Review File · Nature Communications]

ALKBH5 controls the meiosis-coupled mRNA clearance in oocytes by removing the N6-methyladenosine methylationREVIEWER COMMENTS

Reviewer #1 (Remarks to the Author):

Major comments:

The statement "A distinct subset of genes with hyper m6A peaks recruit the m6A reader IGF2BP2 and thus remain stabilized" (Page 2, Line 40-41) is arbitrary due to " genes recruit protein" is unreasonable.

Page 4 line 92-93: The fertility of *Alkbh5*^{-/-} mice had been evaluated in reference (Zheng G et al., Mol Cell. 2013) and *Alkbh5*^{-/-} males were sub-fertile and can produce pups. So the statement that "Male mice with *Alkbh5* deficiency are completely infertile" is subjective.

Page 4 line 96-97: The statement "However, whether ALKBH5 has similar effects in female mice remains unknown." is incorrect. Indeed, the fertility of *Alkbh5*^{-/-} female was also examined in the supplementary Fig.5A of reference "ALKBH5 Is a Mammalian RNA Demethylase that Impacts RNA Metabolism and Mouse Fertility". It is suggested that the authors should compare their results with the reference. Additionally, the number of mice used to test fertility of *Alkbh5*^{-/-} and WT mice should be listed in figure legends or methods.

The authors state that "the impact of temporal m6A dynamics on the GV-to-MII transition remains elusive" may be improper, as this process had been elucidated in 2022 (You Wu et al., Nature Cell Biology. 2022).

The results of the diameters of GV oocytes shown in Fig.S2D is obviously irrational and lack unit.

In Fig. S2G and H, the authors compared the transcriptional difference between *Alkbh5*^{-/-} mice and controls. However, this comparison is inappropriate and unscientific. The authors should take into account the differences in transcription level between the NSN type and SN type oocytes and compare the two different types of oocytes separately.

In Fig. S3C and D, the contents shown in figures are inconsistent with description in results. Are mouse granulosa cells interfered with *Alkbh5* siRNA or from *Alkbh5*^{-/-} mice?

The low-input meRIP-seq method using 2800 oocytes is not advanced, according to the literature (You Wu et al., Nature Cell Biology, 2022). Only about 500 oocytes are needed to get a very solid meRIP-seq result.

The results of RNA immunoprecipitation using granulosa cells could not support the conclusion "ALKBH5 targets maternal transcripts stabilized in *Alkbh5*-null oocytes".

M6A readers are varied and, as mentioned by the author, may be specific. However, the selection strategies of specific readers and the reasons for the authors' focus on IGF2BP2 remain unclear. The authors should compare the combination ability of ALKBH5 with IGF2BP2 and other readers such as YTHDC1/2.

Minor comments:

Page 6 line 168-169: This statement should be modified as "to exclude the effects of mechanical injury"

IHC images of IgG should be shown as negative control in Fig.S1.

Reviewer #2 (Remarks to the Author):

The authors investigate the role of Alkbh5 in oocyte maturation, specifically in maintaining maternal RNA stability through m6A demethylation. The study identifies that depletion of Alkbh5 leads to various defects in oocyte meiosis, resulting in female infertility. Temporal profiling of maternal transcriptomes and m6A dynamics analysis to show that Alkbh5-mediated m6A demethylation is crucial for timely degradation of maternal RNAs. The study also highlights a subset of genes with hyper m6A peaks that recruit the m6A reader Igf2bp2, leading to RNA stabilization and impaired RNA clearance. The authors further demonstrate that reducing Igf2bp2 in Alkbh5-depleted oocytes partially rescues these defects.

The study provides new insights into the role of Alkbh5 in oocyte maturation and emphasizes the significance of m6A demethylation in maternal RNA decay. The authors presented a lot of data, and the results are clearly conveyed. Therefore, this manuscript is recommended for publication in Nat Commun after addressing the following issues.

Major issues:

1. The authors state that the dynamic distribution of Alkbh5 led to the hypothesis that Alkbh5 is required for oocyte maturation. However, it is possible that the observed distribution pattern of ALKBH5 is a consequence of the nuclear envelope breakdown that occurs during oocyte maturation, which could render the nuclear import machinery ineffective. The authors should reconsider this possibility or provide additional evidence to support their hypothesis.
2. In Fig 5A, the authors show the decayed transcripts. It would be helpful to know how many genes were detected in the transcriptome and what percentage of those genes were found to be decayed. Additionally, the authors should clarify whether loss of Alkbh5 affects the degree of decay of the 1400 genes that showed decay in both WT and KO oocytes. Furthermore, it would be interesting to know the expression levels of the extra 410 genes with decay signals in KO oocytes, compared to their expression levels in WT oocytes.
3. The authors are suggested to compare their meRIP-seq data in GV oocytes with at least one of the published studies (PMID: 35606490, 36658155, and 37081317).
4. In Fig 6F, the authors identified 573 genes that might be Alkbh5 regulated m6A+ transcripts. Are these genes upregulated in the temporal RNA profiles of KO oocytes?
5. In Fig 8F, the authors demonstrate that silencing Igf2bp2 in Alkbh5-depleted oocytes partially rescues the defects in oocyte maturation. Do the rescued oocytes have the potential to fertilize and develop?
6. The mechanistic part (Figs 2-3) and profiling part (Figs 4-6) appear to be somewhat disconnected. It would be helpful if the authors could use the dysregulations detected in the profiles to explain the mechanisms behind the observed reductions in MPF activity and inactivation of Cdc20.

Minors:

1. Does Alkbh5 KO affect the expression of Igf2bp2?
2. Regarding the working model (Fig 8H), the authors are recommended to draw out the abnormalities observed in Alkbh5 KO oocytes and show the corresponding percentages.
3. In Fig S1B, the images of the GV oocyte and MII oocyte do not seem to be at the same scale. The authors should check this.
4. Fig 8F lacks scale bars.
5. In line 80-82, the authors should mention an additional recent study (PMID: 37081317) that has revealed the m6A landscape during oocyte maturation. In line 601-602, the authors should also mention two additional recent studies (PMID: 37024923, 37081317) that have explored the regulatory role of m6A in RNA metabolism during MZT.

Reviewer #3 (Remarks to the Author):

Alkbh5 is one of the two known RNA m6A demethylases. Alkbh5^{-/-} male mice are infertile and Alkbh5 is essential for spermatogenesis. This study by Bai, L. et al., further proved that Alkbh5^{-/-} female mice are also infertile and investigated the underlying mechanisms, which largely improves the understanding of roles of Alkbh5 during oogenesis. Oocytes from the Alkbh5^{-/-} female mice are frequently arrested at the MI or GV stages due to severe meiotic maturation defects. The removal of m6A by Alkbh5 on a subset of genes is essential for their timely RNA decay during the GV-to-MII transition. These findings are quite interesting, but this manuscript needs revision before it can be published.

Major points:

The main weakness of this study is that the endogenous substrates of Alkbh5 during oocyte development are unknown. The authors performed RIP-qPCR (but not sequencing) using an antibody specific for Alkbh5 in granulosa cells (but not oocytes) to validate the binding of Alkbh5 to four selected transcripts (Atp5j2, Rpl39, Birc5, and Esrrb). This is not enough to reveal the substrates of Alkbh5 in oocytes. Among the group of 1529 maternal transcripts which are degraded at MII but were stabilized in Alkbh5 KO oocytes, how many of them are indeed targeted and demethylated by Alkbh5? Can the authors perform some kind of low input RNA-RBP interaction study in oocytes to study the substrates of Alkbh5? I know this could be tough. Alternatively, could the author use low-input m6A-DIP-seq datasets in the presence and absence of Alkbh5, as an indirect way to more comprehensively identify Alkbh5 targets?

Another weakness is that the binding targets of IGF2BP2 in oocytes are unknown. This study selected IGF2BP2 just based on the examination of representative transcripts (Atp5j2, Rpl39, Birc5, and Esrrb) which contain CA-rich IGF2BP2 binding motif (lines 666~669). How many A5+m6A+ genes contain the CA-rich IGF2BP2 binding motif? Are the A5+m6A+ genes also containing binding motif for IGF2BP3, IGF2BP1 or some other known m6A readers? How can the authors exclude the possibilities that the A5+m6A+ transcripts are bound by other m6A readers? Whether selected A5+m6A+ transcripts are indeed bound by IGF2BP2 need to be validated by performing low input RNA-RBP interaction study in oocytes.

This study performed low-input m6A meRIP-seq on approximately 2800 GV oocytes from WT and Alkbh5^{-/-} female mice. The effort of collecting so many oocytes for study is appreciated, but there is only one replicate, which is not good for the quality of the dataset. Recently, at least three low-input m6A meRIP-seq datasets in mouse oocytes and early embryos have been published. Can the authors compare their dataset with the published datasets to see whether they are consistent with each other? Can the authors only select those bona fide m6A sites to compare between WT and Alkbh5^{-/-} oocytes, and use these sites for downstream analysis?

Line 571~572: A total of 2450 m6A peaks were differentially changed upon Alkbh5 knockout, with 1325 upregulated and 1125 downregulated enrichments. While Alkbh5 is expected to remove m6A, how does the authors explain the observation of so many (1125 out of 2450) m6A peaks that are downregulated upon Alkbh5 knockout?

This is a newly generated Alkbh5^{+/-} mouse model by CRISPR. What are the phenotypes of the Alkbh5^{-/-} mice? Are they the same as previously reported Alkbh5^{-/-} mouse model? Is the homozygous rate less than 25% in the Alkbh5^{+/-} x Alkbh5^{+/-} mating? Are the size of Alkbh5^{-/-} male or female mice smaller than their Alkbh5^{+/+} and Alkbh5^{+/-} littermates? Although this study focused on oocytes from Alkbh5^{-/-} female mice, the authors should provide more detailed phenotypes of this new mouse line.

The authors were able to collect some MII oocytes from Alkbh5^{-/-} female mice for analysis. Can the Alkbh5 KO MII oocytes be fertilized by WT sperm? If yes, what are the phenotypes of the Alkbh5

maternal KO embryos at the preimplantation stages?

Minor points:

The discussion part is too long, need to be cut and make it concise.

The writing and English of this manuscript could be further polished.

The specificity of antibody is very important. This study used two anti-ALKBH5 antibodies, one for WB, the other one for RIP (Supplementary table S3). Have the authors tested or validated the specificity of these ALKBH5 antibodies? Which anti-ALKBH5 antibody is used for immunohistochemistry in Figure S1A? Which anti-ALKBH5 antibody is used for immunofluorescence to detect the ALKBH5-GFP protein in Figure S1B? Why the authors did not show immunofluorescence results for ALKBH5 in WT oocytes without overexpression? The authors indeed performed immunohistochemistry in *Alkbh5*^{-/-} ovary, but why they did not perform immunofluorescence in *Alkbh5*^{-/-} oocytes?

Line 160~162: Capped mRNAs encoding fluorescently labelled full-length of *Alkbh5* coding region were in vitro transcribed from reconstructed pRK5-Flag-eGFP (a gift from Dr. Hengyu Fan) using mMACHINE™ SP6 kit. Is this pRK5-*Alkbh5*-Flag-eGFP plasmid published before? If yes, please cite the paper. If not, this will be the first time to report this plasmid. The authors should provide the details about the construction this plasmid, such as the primers and method to amplify the full-length *Alkbh5* coding region and cloning strategy.

Line 278~280: The following poly(A) tailing and PCR amplification were performed according to instructions. PCR products were then subjected to enzyme incision for fragmentation. What are the instructions here? What kit did the authors use? What enzyme did authors use for fragmentation? The authors should describe the full details of low input RNA-seq procedure.

1. Summary of major revisions

We thank the editors and reviewers for their positive responses and constructive comments. In light of this feedback, we performed additional analyses and conducted more experiments to improve our manuscript. The revised results are colored in red in our revised manuscript and are summarized as follows:

- (1) We have provided more detailed phenotypes of our *Alkbh5*^{-/-} mice and added the results in **Supplementary Fig. 2c, 2d**.
- (2) We have analyzed the fertilization rate and developmental competence of embryos derived from *Alkbh5*^{-/-} oocytes, and added the results in **Fig. 1l, 1m**.
- (3) We have analyzed the degradation pattern of 1400 genes that showed decay in both WT and *Alkbh5*^{-/-} oocytes and added the results in **Supplementary Fig. 4d, 4e**.
- (4) We have compared our m⁶A-seq data with the other two recently published MeRIP-seq datasets in GV oocytes and displayed results in **Supplementary Fig. 5d-5j**.
- (5) We have narrowed down the scope of ALKBH5 substrates using the combined datasets and identified loss-GMD transcripts as the main substrates of ALKBH5. The results are shown in **Fig. 7**.
- (6) We have compared the combination ability of IGF2BP2 with other readers, confirmed the binding of IGF2BP2 to targeted transcripts in mouse oocytes, and explained our selection strategy. The results are shown in **Fig. 8a-8i**.

2. Point-by-point response to the reviewers' comments

Reviewer #1 (Remarks to the Author):

Major comments:

The statement "A distinct subset of genes with hyper m6A peaks recruit the m6A reader IGF2BP2 and thus remain stabilized" (Page 2, Line 40-41) is arbitrary due to " genes recruit protein" is unreasonable.

Reply:

Thank you for pointing this out. Accordingly, this statement has been revised to "A distinct subset of transcripts with persistent m⁶A peaks are recognized by the m⁶A reader IGF2BP2 and thus remain stabilized" (**Page 2, lines 40-41**).

Page 4 line 92-93: The fertility of *Alkbh5*^{-/-} mice had been evaluated in reference (Zheng G et al., Mol Cell. 2013) and *Alkbh5*^{-/-} males were sub-fertile and can produce pups. So the statement that "Male mice with *Alkbh5* deficiency are completely infertile" is subjective.

Reply:

Thank you for pointing this out. Accordingly, the statement has been revised to "Male mice with *Alkbh5* deficiency are subfertile" (**Page 4, line 93**).

Page 4 line 96-97: The statement "However, whether ALKBH5 has similar effects in female mice remains unknown." is incorrect. Indeed, the fertility of *Alkbh5*^{-/-} female was also examined in the supplementary Fig.5A of reference "ALKBH5 Is a Mammalian RNA Demethylase that Impacts RNA Metabolism and Mouse Fertility". It is suggested that the authors should compare their results with the reference. Additionally, the number of mice used to test fertility of *Alkbh5*^{-/-} and WT mice should be listed in figure legends or methods.

Reply:

We thank the reviewer for identifying this error. Accordingly, the statement has been revised to “Aside from its roles in male fertility, ALKBH5 appears to be involved in the regulation of oocyte development, as female *Alkbh5*^{-/-} mice also show impaired fertility. However, it remains unknown how ALKBH5 affects female fertility and whether the demethylation function of ALKBH5 is involved” (**Page 4, lines 96-99**).

The previous *Alkbh5*^{-/-} mouse line was established by LoxP/Cre-mediated excision of *Alkbh5*, as mentioned by the reviewer (PMID: 23177736). However, in our present study, a new *Alkbh5* knockout mouse strain was generated by ablating *Alkbh5* exon 1 using the CRISPR-Cas9 system. To assess the fertility of these mice, we set up three pairs of matings, each involving one 2-month-old *Alkbh5*^{-/-} female and one wild-type male, for more than 150 days of continuous breeding. Notably, none of the *Alkbh5*^{-/-} females had litters. Complete infertility was also supported by the total absence of 4-cell embryos derived from KO oocytes at the corresponding time. We did notice that Zheng et al. showed a remarkably low but detectable success rate of breeding (1.46 pups per mating for a total of 26 matings) when *Alkbh5*^{-/-} females were crossed with heterozygous males. We believe that this could be justified by the difference in generation methods, female numbers for mating and breeding duration. This point has been discussed in the revised manuscript (**Page 28, lines 818-827**).

Additionally, we listed the number of mice used to test the fertility of *Alkbh5*^{-/-} and WT mice in both the figure legends (**Pages 35-36, lines 1050-1051**) and methods (**Page 4, lines 118-119**).

The authors state that "the impact of temporal m⁶A dynamics on the GV-to-MII transition remains elusive" may be improper, as this process had been elucidated in 2022 (You Wu et al., Nature Cell Biology. 2022).

Reply:

We thank the reviewer for the suggestion. Accordingly, the statement has been revised. Please see **Page 22, lines 646-648**.

The results of the diameters of GV oocytes shown in Fig.S2D is obviously irrational and lack unit.

Reply:

We thank the reviewer for the kind suggestion. Oocyte morphology and diameter are correlated with oocyte quality, and poor morphology with a smaller diameter was observed in oocytes with impaired oocyte growth (for example: Fig. 4 of PMID: 30283081). Thus, examination of the diameters and morphology of GV oocytes has been used as the overall assessment of oocyte growth in preceding studies (for example: Fig. S1 of PMID: 32313933; Fig. S1 of PMID: 29408237). In addition, oocytes with depletion of the m⁶A writer *Kiaa1429* (PMID: 32094512) were smaller in size with abnormal cytoplasmic granules, as were *Ythdc1*-depleted oocytes (PMID: 29799838). Moreover, *Mettl3*-null oocytes also had a smaller GV diameter (PMID: 34689175), suggesting that m⁶A may contribute to morphologic defects for unknown reasons.

For the above two reasons, we believe that the calculation of GV diameters here is quite reasonable considering the possible impact of *Alkbh5* loss on oocyte growth. However, we apologize that Fig.

S2D in the original manuscript lacked units; we have resolved this error (**Revised Supplementary Fig. 2f**).

In Fig. S2G and H, the authors compared the transcriptional difference between *Alkbh5*^{-/-} mice and controls. However, this comparison is inappropriate and unscientific. The authors should take into account the differences in transcription level between the NSN type and SN type oocytes and compare the two different types of oocytes separately.

Reply:

We thank the reviewer for the constructive comments. We have already compared the ratios of oocytes that reached the SN and NSN configuration between WT and KO oocytes (**Revised Supplementary Fig. 2g, 2h**). Given that a similar proportion of transcriptionally active and NSN oocytes was observed (33.0% NSN versus 32.0% transcriptionally active in WT oocytes, 73.8% NSN versus 77.4% transcriptionally active in KO oocytes), we believe that the observed transcriptional silencing defect is due to the gross defect in chromatin organization. Thus, we deleted the figures as pointed out the reviewer.

In Fig. S3C and D, the contents shown in figures are inconsistent with description in results. Are mouse granulosa cells interfered with *Alkbh5* siRNA or from *Alkbh5*^{-/-} mice?

Reply:

Thank you for pointing this out. We apologize for the incorrect description in the results. Accordingly, the description has been revised to “Similar results were also detected in mouse granulosa cells in which endogenous *Alkbh5* expression was inhibited by the corresponding siRNA, which suggested that ALKBH5 might also be involved in mitosis” (**Page 17, lines 483-485**).

The low-input meRIP-seq method using 2800 oocytes is not advanced, according to the literature (You Wu et al., *Nature Cell Biology*, 2022). Only about 500 oocytes are needed to get a very solid meRIP-seq result.

Reply:

Thank you for this point. We fully agree with the reviewer’s comment that our m⁶A-seq using 2800 oocytes “is not advanced”, as the ULI-MeRIP-seq mentioned by the reviewer had not been reported until we finished our sequencing.

During the preparation of our manuscript, several advanced m⁶A sequencing techniques were developed and reported, which enabled the profile of m⁶A RNA methylation with only 50 oocytes (PMID:37081317) or even a single oocyte (PMID:36658155). We compared our m⁶A-seq data with these datasets and found that nearly 86% of m⁶A-tagged transcripts detected by picoMeRIP-seq and approximately 68% by scm⁶A-seq were also identified in GV oocytes in our dataset using bulk MeRIP-seq (**Rebuttal Fig. 1a and Revised Supplementary Fig. 5d**). Additionally, the 2110 transcripts detected by all three different mapping techniques were greatly enriched for critical biological functions (**Rebuttal Fig. 1b and Revised Supplementary Fig. 5e**), which was consistent with previous findings; we provided example IGV snapshots for some of these genes (**Rebuttal Fig. 1c and Revised Supplementary Fig. 5f**).

Collectively, these results validated the reproducibility and quality of our m⁶A-seq data.

Rebuttal Fig 1. Comparative analysis of m⁶A-seq datasets.

a. Venn diagram showing the overlaps of identified m⁶A-modified RNAs in GV oocytes from picoMeRIP-seq, scm⁶A-seq and our datasets.

b. GO analysis of commonly identified m⁶A-marked RNAs by different methods. Pos. positive; neg. negative; reg. regulation.

c. IGV snapshots showing m⁶A distributions of the representative conserved m⁶A-modified genes (*Mos*, *Mastl*, *Cebpb*, *Foxo1*) detected by different methods. The sample-specific scale ranges are indicated in brackets.

The results of RNA immunoprecipitation using granulosa cells could not support the conclusion “ALKBH5 targets maternal transcripts stabilized in Alkbh5-null oocytes” .

Reply:

Thank you for pointing this out. We performed RNA immunoprecipitation of these candidate transcripts (*Rpl39*, *Atp5j2*, *Esrrb*, *Birc5*) for ALKBH5 in **GV oocytes** three times and showed that ALKBH5 enrichment at these transcripts was markedly higher than that of IgG (**Rebuttal Fig. 2a and Revised Fig. 7m-7p**). Thus, we validated the binding of candidate transcripts to ALKBH5 in mouse oocytes.

Rebuttal Fig 2. ALKBH5 RIP results in mouse oocytes.

a. RIP-qPCR analysis showing the fold changes of ALKBH5 binding ability with the indicated RNAs relative to IgG in mouse oocytes. The bar is mean \pm SD resulting from three independent

experiments. P value, unpaired t test.

Additionally, in the revised manuscript, we narrowed the scope and identified the major substrates of ALKBH5 (please see detailed results in **lines 674-736, Revised Fig. 7**). Specifically, in Fig. 7, we grouped all m⁶A⁺ GMD transcripts into two clusters according to their m⁶A dynamics during the GV-MII transition. A subset of m⁶A⁺ transcripts that had lost m⁶A modification by the MII stage (termed “loss-GMD”) was of particular interest since these transcripts showed greater degradation than the inherited-GMD transcripts. However, this degradation difference was compromised upon *Alkbh5* knockout. Thus, we reasoned that genes that have lost m⁶A (loss-GMD) are the major substrates of ALKBH5.

To this end, we confirmed the m⁶A loss of candidate transcripts in MII oocytes and showed that *Alkbh5* depletion led to persistent m⁶A modification at the MII stage. In addition to persistent m⁶A status, the degradation of loss-GMD transcripts was depressed upon *Alkbh5* knockout (some transcripts were stabilized or even upregulated by the MII stage), suggesting that these transcripts tended to be stabilized upon *Alkbh5* depletion. Moreover, we validated the binding of candidate transcripts to ALKBH5, as mentioned above.

Therefore, we believe that these new data are sufficient to support our conclusion that “ALKBH5 mainly targets loss-GMD transcripts in mouse oocytes”. We have clarified this point in **Pages 23-25, lines 674-736** and added multiple new evidence to strengthen our idea in the main figure (**Revised Fig. 7**).

M6A readers are varied and, as mentioned by the author, may be specific. However, the selection strategies of specific readers and the reasons for the authors' focus on IGF2BP2 remain unclear. The authors should compare the combination ability of ALKBH5 with IGF2BP2 and other readers such as YTHDC1/2.

Reply:

Thank you for your helpful suggestions. As the reviewer noted, there are various types of m⁶A readers, and they guide diverse functions. Some of them are associated with RNA stability in mouse oocytes and embryos, as demonstrated in previous studies. An impressive study from the Ivanova lab published in 2017 (PMID: 28867294) first established the destabilizing effect of YTHDF2 on maternal RNAs in mouse oocytes by showing a deregulated *Ythdf2*-cKO MII transcriptome with gene expression bias toward upregulation. The regulatory role of YTHDF2 on RNA decay after fertilization, but not during oocyte maturation, was further emphasized by Wu et al (PMID: 35606490). In contrast to the degradation function of YTHDF2, IGF2BPs, another well-known family of m⁶A readers, were reported to guard m⁶A-modified RNAs from decay (PMID: 29476152), and IGF2BP3 participates in Mettl3-mediated RNA stabilization in mouse oocytes (PMID: 34689175). In addition to IGF2BPs, a very recent study has indicated that YTHDC1 is a regulator in maintaining RNA stability in mouse embryos (PMID: 37024923). Thus, we reasoned that one of these readers is the likely mediator of the RNA decay defect observed in *Alkbh5* KO oocytes.

To determine the specific reader involved, we first analyzed the binding sites of these readers from public RIP-seq and eCLIP-seq datasets derived from mouse and human ESCs, and the distance

between m⁶A peaks of loss-GMD (major substrates of ALKBH5) and binding peaks was calculated. We found that m⁶A-modified regions were significantly accessible to IGF2BP2 and 3, especially for IGF2BP2 (**Rebuttal Fig. 3a and Revised Fig. 8a**) (IGF2BP1 was excluded due to its relatively low expression level). **As requested by the reviewer, we also compared the ability of candidate gene transcripts to bind to these readers in mouse GV oocytes** (YTHDC2 was not tested since there is no existing evidence showing its relationship with RNA stability). The results consistently showed that the binding to IGF2BP2 was markedly higher than those of any other readers (**Rebuttal Fig. 3b-3e and Revised Fig. 8b-8e**). Furthermore, we investigated whether IGF2BP2 binds to these transcripts when oocytes reach the **MII stage**. As expected, IGF2BP2 enrichment at these transcripts was disrupted in WT MII oocytes but maintained at a relatively high level in KO MII oocytes (**Rebuttal Fig. 3f-3i and Revised Fig. 8f-8i**), which clearly supported our proposed model that the m⁶A enrichment of these transcripts abnormally persisted in KO MII oocytes (**Revised Fig. 7i**).

Therefore, we considered IGF2BP2 to be the main m⁶A reader involved, and this hypothesis was further supported by rescue experiments with microinjection of *siIgf2bp2*.

Rebuttal Fig 3. IGF2BP2 are the main reader involved in this process.

a. The density plot showing the distance between binding sites of m⁶A readers (IGF2BP2, IGF2BP3, YTHDF2 and YTHDC1) detected by RIP-seq and eCLIP-seq in mouse and human ESCs and m⁶A peaks of loss-GMD transcripts detected by m⁶A-seq in our dataset.

b-e. RIP-qPCR results showing the relative enrichments of the indicated m⁶A readers among the targeted genes in mouse oocytes. The data validated by qPCR are present as mean ±SD and are normalized to IgG samples. P value, unpaired t test.

f-i. RIP-qPCR results showing the relative enrichments of IGF2BP2 among the targeted genes in WT and *Alkbh5*^{-/-} MII oocytes. The data validated by qPCR are present as mean ±SD and are normalized to IgG samples. P value, unpaired t test.

Minor comments:

Page 6 line 168-169: This statement should be modified as "to exclude the effects of mechanical injury"

Reply:

Thank you for pointing this out. We have revised the manuscript accordingly (**Page 7, lines 191-192**).

IHC images of IgG should be shown as negative control in Fig.S1.

Reply:

We thank the reviewer for the constructive comments. We repeated the ALKBH5 IHC experiments with IgG as a negative control simultaneously and updated Fig. S1 (**Rebuttal Fig. 4 and Revised Supplementary Fig. 1a**). The picture below clearly shows the nuclear localization of ALKBH5 in mouse oocytes.

a

Rebuttal Fig 4. ALKBH5 IHC results of ovarian section.

a. Immunohistochemistry results showing ALKBH5 location in oocyte during follicle development. The dark arrow indicates the primordial follicle enveloped by a layer of squamous cells and asterisk highlights the primary follicle. Immunohistochemistry images of IgG are shown as negative control.

Reviewer #2 (Remarks to the Author):

The authors investigate the role of Alkbh5 in oocyte maturation, specifically in maintaining maternal RNA stability through m6A demethylation. The study identifies that depletion of Alkbh5 leads to various defects in oocyte meiosis, resulting in female infertility. Temporal profiling of maternal transcriptomes and m6A dynamics analysis to show that Alkbh5-mediated m6A demethylation is crucial for timely degradation of maternal RNAs. The study also highlights a subset of genes with hyper m6A peaks that recruit the m6A reader Igf2bp2, leading to RNA stabilization and impaired RNA clearance. The authors further demonstrate that reducing Igf2bp2 in Alkbh5-depleted oocytes partially rescues these defects.

The study provides new insights into the role of Alkbh5 in oocyte maturation and emphasizes the significance of m6A demethylation in maternal RNA decay. The authors presented a lot of data, and the results are clearly conveyed. Therefore, this manuscript is recommended for publication in Nat Commun after addressing the following issues.

Major issues:

1. The authors state that the dynamic distribution of Alkbh5 led to the hypothesis that Alkbh5 is required for oocyte maturation. However, it is possible that the observed distribution pattern of ALKBH5 is a consequence of the nuclear envelope breakdown that occurs during oocyte maturation, which could render the nuclear import machinery ineffective. The authors should reconsider this possibility or provide additional evidence to support their hypothesis.

Reply:

Thank you for making this point clear. Indeed, we **could not exclude the possibility** that cytoplasmic redistribution of ALKBH5 is the consequence of ineffective nuclear import after GVBD. Regarding ALKBH5 localization, few studies have focused on mouse oocytes, and to our knowledge, the current study represents the first observation of ALKBH5 cytoplasmic localization in MII oocytes. Furthermore, we noticed a similar distribution pattern of METTL3 (m⁶A writer) during oocyte maturation, while the mechanism of METTL3 distribution pattern remained unclear (PMID: 34689175). Therefore, at this point, we have no clear explanation for this phenomenon.

To avoid misinterpretation, we have modified our statement as follows: “To further examine the *in vivo* function of this factor, we generated *Alkbh5* total knockout mice by ablating exon 1, including the start codon, from the *Alkbh5* gene using the CRISPR-Cas9 system” (**Page 14, lines 416-418**).

2. In Fig 5A, the authors show the decayed transcripts. It would be helpful to know how many genes were detected in the transcriptome and what percentage of those genes were found to be decayed. Additionally, the authors should clarify whether loss of Alkbh5 affects the degree of decay of the 1400 genes that showed decay in both WT and KO oocytes. Furthermore, it would be interesting to know the expression levels of the extra 410 genes with decay signals in KO oocytes, compared to their expression levels in WT oocytes.

Reply:

Thank you for the helpful suggestions. A total of 33090 genes annotated in the ENSEMBL gene annotation library were obtained in our transcriptome. To exclude the genes with relatively low RNA transcript levels, we identified **13278** expressed genes (FPKM >1 in at least one sample). Among the expressed genes, approximately **22.1% (2929/13278)** showed transcript degradation in WT oocytes, while only **13.6% (1810/13278)** showed transcript degradation in *Alkbh5* KO oocytes.

Regarding the impact of *Alkbh5* loss on the degree of decay of the 1400 transcripts, we performed additional analyses. We compared the extent and rate of degradation of these transcripts between WT and KO oocytes. As presented below, ***Alkbh5* loss also depressed the degradation of these transcripts that showed decay in both WT and KO oocytes (Rebuttal Fig. 5a, 5b and Revised Supplementary Fig. 4c, 4d)**, suggesting a global role for ALKBH5 in destabilizing the overall transcriptome. We have clarified this point in the revised manuscript (**Page 20, lines 594-597**) and added the related data to the Supplementary Figures.

Furthermore, as requested, we also showed the relative expression levels of the extra 410 genes in WT and KO oocytes from the GV to MII stage, as illustrated by a heatmap (**Rebuttal Fig. 5c**). Intriguingly, compared to the controls, **these genes exhibited a much higher RNA abundance at the GV stage but almost comparable RNA levels at the MII stage in KO oocytes**, suggesting that these exclusively decayed transcripts tend to be more abundant in the GV stage upon *Alkbh5*

knockout.

Rebuttal Fig 5. The impact of *Alkbh5* loss on the degree of decay of maternal transcripts.

a. Boxplot depicting the \log_2 [fold change (MII/GV)] of 1400 genes that degrade in both WT and *Alkbh5*^{-/-} oocytes compared between WT and KO samples.

b. Degradation patterns of maternal transcripts (1400 genes) during GV-MII transition in WT and *Alkbh5*^{-/-} oocytes. Each green line represents the expression level of an individual gene. The blue and red line represent the median expression levels in WT and *Alkbh5*^{-/-} oocyte, respectively.

c. Heatmap illustrating the relative expression levels of the extra 410 genes in WT and KO oocytes from the GV to MII stage.

3. The authors are suggested to compare their meRIP-seq data in GV oocytes with at least one of the published studies (PMID: 35606490, 36658155, and 37081317).

Reply:

Thank you for the suggestion. As suggested by the reviewer, we analyzed picoMeRIP-seq data for 49 GV oocytes with two replicates (PMID: 37081317) and aggregated scm⁶A-seq data of single GV oocytes with ten replicates (PMID: 36658155). We compared them with our bulk m⁶A-seq data from 2800 GV oocytes filtered using the same criteria.

The Venn diagram shown below clearly revealed that m⁶A-modified transcripts detected by our m⁶A-seq data covered approximately 85.5% of m⁶A-modified transcripts from picoMeRIP-seq data, approximately 68.1% of merged scm⁶A-seq data and nearly all (93.0%) transcripts that were identified as modified in both datasets (**Rebuttal Fig. 1a and Revised Supplementary Fig. 5d**). We then performed GO analysis of the commonly modified transcripts that were present in all three datasets. The results showed that these genes were greatly involved in critical biological pathways in GV oocytes, including the regulation of transcription, chromatin organization and the cell cycle, which is consistent with a previous study (PMID: 37081317, PMID: 36658155) (**Rebuttal Fig. 1b and Revised Supplementary Fig. 5e**). Moreover, the key genes for the preparation for meiotic resumption in GV oocytes were m⁶A-modified and highly conserved among three datasets, such as *Mos* and *Mastl*, which are required for cell cycle progression, and *Cebpb* and *Foxo1*, which are required for transcription (**Rebuttal Fig. 1c and Revised Supplementary Fig. 5f**). We have added the comparison figures and results to our revised manuscript (**Pages 22-23, lines 659-673**).

Rebuttal Fig 1. Comparative analysis of m⁶A-seq datasets.

- a.** Venn diagram showing the overlaps of identified m⁶A-modified RNAs in GV oocytes from picoMeRIP-seq, scm⁶A-seq and our datasets.
- b.** GO analysis of commonly identified m⁶A-marked RNAs by different methods. Pos. positive; neg. negative; reg. regulation.
- c.** IGV snapshots showing m⁶A distributions of the representative conserved m⁶A-modified genes (*Mos*, *Mastl*, *Cebpb*, *Foxo1*) detected by different methods. The sample-specific scale ranges are indicated in brackets.

4. In Fig 6F, the authors identified 573 genes that might be Alkbh5 regulated m⁶A+ transcripts. Are these genes upregulated in the temporal RNA profiles of KO oocytes?

Reply:

Thank you for this comment. We compared the RNA abundance of these 573 genes between WT and KO oocytes at the MII stage and found that **they were indeed increased upon *Alkbh5* knockout (Rebuttal Fig. 6).**

Rebuttal Fig 6. RNA abundance of A⁵⁺m⁶A⁺ transcripts at MII stage.

- a.** Violin plot depicting the relative RNA abundance of A⁵⁺m⁶A⁺ transcripts at MII stage compared between WT and KO samples.

Furthermore, to determine the bona fide ALKBH5 targets, we focused on the transcripts that were degraded during the GV-MII transition in WT (136 genes) and observed modestly depressed

degradation upon *Alkbh5* depletion. However, only a small number of these transcripts (65 genes) were stabilized (unchanged or upregulated) in KO MII oocytes, suggesting that A5⁺m⁶A⁺ transcripts account for only part of the observed decay defect (**Revised Supplementary Fig. 6e-6i**). Given that the KO GV oocytes contained a near normal transcriptome (179 decreased, 244 increased), we reasoned that ALKBH5 may have a limited effect on RNA decay **before the GV stage**.

To address this issue, we further expanded the analysis by focusing on the possibility that ALKBH5 functions **after the GV stage** (that is, after the occurrence of RNA decay). We identified loss-GMD transcripts as major substrates of ALKBH5 and confirmed their persistent m⁶A enrichment and upregulated RNA abundance in MII KO oocytes and validated their binding to ALKBH5 in mouse oocytes. Please find detailed results in **lines 674-736, Revised Fig. 7**.

5. In Fig 8F, the authors demonstrate that silencing *Igf2bp2* in *Alkbh5*-depleted oocytes partially rescues the defects in oocyte maturation. Do the rescued oocytes have the potential to fertilize and develop?

Reply:

Thank you for making this point clear. As shown in Fig. 8F in the original manuscript, we observed a partially rescued but still greatly decreased PBE rate (from ~10% of KO to ~30% of KO with *Igf2bp2* knockdown). These findings prompted us to hypothesize that rescued oocytes may have the potential to be fertilized and develop. *In vitro* fertilization of the rescued oocytes and embryo culture could be one experiment performed to test this hypothesis. However, the long duration of microinjection experiments (12 h for the RNAi effect and another 18 h for maturation) further attenuates KO oocyte competence, which has already been impaired by *Alkbh5* loss. **Thus, it was technically difficult to acquire adequate mature oocytes to assess fertilization ability in our case.** Ideally, the generation of an *Igf2bp2* and *Alkbh5* double knockout mouse line could be used for this purpose to avoid interference from microinjection. However, such experiments would be beyond the scope of our current paper.

As an alternative, we tested the fertilization capacity and monitored subsequent embryo development in *Alkbh5* KO females. The figure below revealed that approximately half of the ovulated KO oocytes could be fertilized as characterized by pronucleus formation 28 h after hCG injection. After fertilization, we selected fertilized oocytes from the oviducts for *in vitro* culture. Only approximately 40% of embryos derived from KO females reached the 2-cell stage, and none developed beyond this point (**Rebuttal Fig. 7a, 7b and Revised Fig. 1l, 1m**), reflecting the compromised developmental potential of KO oocytes.

Therefore, we reasoned that the rescued oocytes with *Igf2bp2* knockdown are likely to be fertilized but are ultimately arrested before the 4-cell stage, even though they showed improved oocyte quality.

Rebuttal Fig 7. Embryos derived from *Alkbh5*^{-/-} oocytes were arrested at the 2-cell stage.

a. Representative pictures of embryos derived from WT and *Alkbh5*^{-/-} oocytes at 28, 40 and 54 h after hCG injection.

b. Developmental rates of embryos that reach the indicated stages at corresponding timepoints. The data are presented as mean \pm SD. P value, unpaired t test.

6. The mechanistic part (Figs 2-3) and profiling part (Figs 4-6) appear to be somewhat disconnected. It would be helpful if the authors could use the dysregulations detected in the profiles to explain the mechanisms behind the observed reductions in MPF activity and inactivation of Cdc20.

Reply:

Thank you for the helpful suggestions. As noted by the reviewer, in Fig. 3, we have demonstrated that APC/CDC20 inactivation is partly due to prolonged SAC activation and thus contributes to MI arrest in KO oocytes (as shown by the persistent kinetochore localization of BubR1 at the AI stage). Of note, the SAC is known as a surveillance system that coordinates kinetochore-microtubule attachment and anaphase onset (PMID: 26485365). Its inhibitory signal is abrogated only after all chromosomes have achieved the correct attachment to the opposite spindle poles.

Focusing on master regulators essential for spindle and chromosome events, we observed accumulation of transcripts corresponding to certain genes in KO MII oocytes. The key genes for the correction of erroneous kinetochore-microtubule attachment in mouse oocytes were upregulated, including *Zwint* and *Cdk7*. The association between dysregulation of these proteins and impaired SAC has been reported in previous studies (PMID: 27693251 and 26486467). RNA transcripts of genes regulating chromosome kinetics, such as *Birc5*, which is required for chromosome segregation (PMID: 30382773), and *Septin2*, which is required for chromosome congression (PMID: 20372094), also accumulated. Interestingly, *Birc5* (which encodes the Survivin protein), which has been reported as a critical chromosomal passenger complex (CPC) subunit, showed apparent m⁶A modification in GV oocytes and was proven to be subjected to ALKBH5-mediated stabilization in our follow-up experiments. Moreover, elevated RNA levels of *Ska3*, a member of the spindle and kinetochore-associated complex, were also observed in KO MII oocytes; its role in controlling spindle stability has been well studied (PMID:22336914).

In addition, among these accumulated transcripts, CDC25B is of particular interest, as it is well established as a key phosphatase that mediates CDK1 dephosphorylation and thus governs cell cycle progression. It is maintained at a relatively low level in GV oocytes but greatly increased after meiotic resumption and ultimately decreased by the MI stage. CDC25 accumulation could

contribute to spindle and chromosome abnormalities in mouse oocytes (PMID: 26626423). Based on our RNA-seq dataset, the *Cdc25b* RNA levels in KO oocytes were comparable to those in WT oocytes until GVBD. However, *Cdc25b* was observed with a markedly higher expression level in KO MII oocytes.

Overall, we reasoned that the accumulation of these transcripts is the likely cause of spindle and chromosome defects in KO oocytes, which activate the SAC and thus contribute to inactivate APC/CDC20, resulting in MI arrest in KO oocytes.

We have added this part in the Discussion section (Pages 28-29, lines 828-856).

Minors:

1. Does *Alkbh5* KO affect the expression of *Igf2bp2*?

Reply:

Thank you for making this point clear. Comparative analysis with the RNA-seq dataset revealed that the RNA levels of *Igf2bp2* in KO oocytes were comparable to those in WT oocytes (**Rebuttal Fig. 8**), suggesting that *Alkbh5* depletion did not affect the expression of *Igf2bp2* at the RNA level.

Rebuttal Fig 8. *Alkbh5* depletion did not affect the expression of *Igf2bp2* at the RNA level.

a. Bar plot showing the relative expression fold changes of *Igf2bp2* upon *Alkbh5* knockout.

2. Regarding the working model (Fig 8H), the authors are recommended to draw out the abnormalities observed in *Alkbh5* KO oocytes and show the corresponding percentages.

Reply:

We thank the reviewer for the constructive comments. We have modified our working model accordingly (**Revised Fig. 9**).

3. In Fig S1B, the images of the GV oocyte and MII oocyte do not seem to be at the same scale. The authors should check this.

Reply:

Thank you for the kind suggestion. We have reviewed the figure as mentioned by the reviewer and have added a white dashed circle to outline the oocyte (**Revised Supplementary Fig. 1b**).

4. Fig 8F lacks scale bars.

Reply:

Thank you for pointing this out. We have fixed the mistake (**Revised Fig. 8m**).

5. In line 80-82, the authors should mention an additional recent study (PMID: 37081317) that has revealed the m6A landscape during oocyte maturation. In line 601-602, the authors should also mention two additional recent studies (PMID: 37024923, 37081317) that have explored the regulatory role of m6A in RNA metabolism during MZT.

Reply:

Thank you for pointing this out. We have cited the mentioned references (PMID: 37081317, PMID: 37024923) in the corresponding text (**Pages 3, lines 80-82; Pages 22, lines 646-648**).

Reviewer #3 (Remarks to the Author):

Alkbh5 is one of the two known RNA m6A demethylases. Alkbh5^{-/-} male mice are infertile and Alkbh5 is essential for spermatogenesis. This study by Bai, L. et al., further proved that Alkbh5^{-/-} female mice are also infertile and investigated the underlying mechanisms, which largely improves the understanding of roles of Alkbh5 during oogenesis. Oocytes from the Alkbh5^{-/-} female mice are frequently arrested at the MI or GV stages due to severe meiotic maturation defects. The removal of m6A by Alkbh5 on a subset of genes is essential for their timely RNA decay during the GV-to-MII transition. These findings are quite interesting, but this manuscript needs revision before it can be published.

Major points:

The main weakness of this study is that the endogenous substrates of Alkbh5 during oocyte development are unknown. The authors performed RIP-qPCR (but not sequencing) using an antibody specific for Alkbh5 in granulosa cells (but not oocytes) to validate the binding of Alkbh5 to four selected transcripts (Atp5j2, Rpl39, Birc5, and Esrrb). This is not enough to reveal the substrates of Alkbh5 in oocytes. Among the group of 1529 maternal transcripts which are degraded at MII but were stabilized in Alkbh5 KO oocytes, how many of them are indeed targeted and demethylated by Alkbh5? Can the authors perform some kind of low input RNA-RBP interaction study in oocytes to study the substrates of Alkbh5? I know this could be tough. Alternatively, could the author use low-input m6A-DIP-seq datasets in the presence and absence of Alkbh5, as an indirect way to more comprehensively identify Alkbh5 targets?

Reply:

We thank the reviewer for the constructive comments. As suggested by the reviewer, we combined our RNA-seq datasets from WT and KO oocytes at GV and MII stages with public picoMeRIP-seq data from mouse oocytes at the GV to MII stages (PMID: 37081317) and performed additional analyses to identify ALKBH5 substrates. **Here, we propose two scenarios for how *Alkbh5* deficiency leads to RNA degradation defects in KO oocytes and confirmed the major substrates of ALKBH5 in the following experiments.** Please find the schematics below showing the two scenarios (**Rebuttal Fig. 9a, 9b**).

Rebuttal Fig 9. Working models of ALKBH5 for maternal RNA decay.

a-b. The schematics showing two scenarios for how *Alkbh5* deficiency leads to RNA degradation defects in KO oocytes.

First, **during the GV-MII transition**, ALKBH5 removes m^6A from a subset of transcripts, thereby promoting timely RNA decay. Thus, *Alkbh5* depletion results in the maintenance of m^6A on these transcripts at MII stage, which in turn compromises their degradation. We identified 298 transcripts as “loss-GMD”, which harbored m^6A modification at the GV stage but lost m^6A by the MII stage. In support of this notion, we observed that m^6A is preferentially lost from transcripts that are degraded during oocyte maturation and that transcripts that have lost m^6A degrade more rapidly than those with unchanged m^6A status. However, this disparity in degradation was reduced upon *Alkbh5* depletion.

Furthermore, we conducted MeRIP-qPCR of candidate genes (*Rpl39*, *Birc5*, *Atp5j2* and *Esrrb*) in mouse oocytes and confirmed m^6A loss from the GV to MII stage in WT oocytes and persistent m^6A enrichment in KO MII oocytes. Next, we observed a massive decrease in the degradation of loss-GMD transcripts: approximately 79.5% of these 298 transcripts were affected (KO(MII/GV)/WT(MII/GV)>1), of which approximately 67.0% remained stabilized or even upregulated in KO MII oocytes. Moreover, we validated the binding of these transcripts to ALKBH5 in mouse oocytes. **Thus, we concluded that loss-GMD transcripts are indeed the major substrates of ALKBH5 during the GV-MII transition.** (Please see detailed results in **lines 674-736** and **Revised Fig. 7**).

Second, ALKBH5 may function **at the early phase of oocyte growth** (before the occurrence of RNA decay), and the resultant non- m^6A -modified transcripts undergo degradation after this point. Thus, *Alkbh5* depletion results in the exclusive gain of m^6A in a subset of transcripts at the GV stage, which in turn causes the observed defects. However, among the decayed transcripts, only a small number (136 genes) were detected with exclusive m^6A in KO oocytes. Indeed, these transcripts exhibited a milder depression of degradation upon *Alkbh5* knockout, and 65 of them were stabilized in KO MII oocytes. Thus, we reasoned that scenario 2 could account for only a part of the impaired RNA decay in KO oocytes (Please see detailed results in **lines 737-754** and **Revised Supplementary Fig. 6e-6k**).

Overall, we further defined loss-GMD transcripts as major substrates of ALKBH5 with sufficient evidence and validated candidate genes in scenario 1 in mouse oocytes. We have added these new data to **Revised Fig. 7** and **Supplementary Fig. 6e-6k** and modified the manuscript accordingly (**Pages 23-26, lines 674-754**).

Another weakness is that the binding targets of IGF2BP2 in oocytes are unknown. This study selected IGF2BP2 just based on the examination of representative transcripts (Atp5j2, Rpl39, Birc5, and Esrrb) which contain CA-rich IGF2BP2 binding motif (lines 666~669). How many A5+m6A+ genes contain the CA-rich IGF2BP2 binding motif? Are the A5+m6A+ genes also containing binding motif for IGF2BP3, IGF2BP1 or some other known m6A readers? How can the authors exclude the possibilities that the A5+m6A+ transcripts are bound by other m6A readers? Whether selected A5+m6A+ transcripts are indeed bound by IGF2BP2 need to be validated by performing low input RNA-RBP interaction study in oocytes.

Reply:

Thank you for the helpful comments. Although m⁶A readers are from different families and mediate various functions, certain readers have been shown to be associated with RNA stability in previous studies, including YTHDF2, YTHDC1 and IGF2BPs. Here, IGF2BP1 was excluded due to its relatively low RNA levels in mouse oocytes (FPKM=0 in our dataset). To determine the bona fide reader involved and exclude other possibilities, we performed additional analyses and experiments as follows:

- 1) We analyzed the binding sites of the m⁶A readers mentioned above in the public RIP-seq and eCLIP-seq datasets conducted in mESCs and hESCs and compared them with the m⁶A peaks of the ALKBH5 substrates identified above (that is, loss-GMD). We found that the targets of only two readers, IGF2BP2 and IGF2BP3 (especially IGF2BP2), significantly overlapped with the m⁶A signatures of these 298 genes when compared to the random background, indicating that **m⁶A regions of loss-GMD are easily accessible to these two readers (Rebuttal Fig. 3a and Revised Fig. 8a).**
- 2) We then compared the binding ability of these readers to candidate transcripts **in GV oocytes** and obtained consistent results. The binding of IGF2BP2 to these transcripts is significantly higher than that of IgG (IgG is normalized to 1) and, notably, **is markedly higher than that of any other reader (Rebuttal Fig. 3b-3e and Revised Fig. 8b-8e).**
- 3) We compared the binding of IGF2BP2 to candidate transcripts **in WT and KO oocytes at the MII stage**. We found that **IGF2BP2 enrichment at these transcripts was disrupted in WT MII oocytes; in contrast, IGF2BP2 maintained its binding ability in KO MII oocytes (Rebuttal Fig. 3f-3i and Revised Fig. 8f-8i).** This result is in accordance with the notion that loss-GMD transcripts fail to lose m⁶A in KO MII oocytes (scenario 1).

In short, these new results clearly justified our selection strategy and validated the binding of IGF2BP2 to loss-GMD transcripts.

However, we could not rule out the possibility that a small number of transcripts may be bound by other readers, since this process is complicated and still not well understood. Thus, we have changed the sentence in the main text to emphasize IGF2BP2 as the major (but not the only) reader involved and mentioned this possibility in the Discussion section (**Pages 30, lines 872-892**).

Rebuttal Fig 3. IGF2BP2 are the main reader involved in this process.

a. The density plot showing the distance between binding sites of m^6A readers (IGF2BP2, IGF2BP3, YTHDF2 and YTHDC1) detected by RIP-seq and eCLIP-seq in mouse and human ESCs and m^6A peaks of loss-GMD transcripts detected by m^6A -seq in our dataset.

b-e. RIP-qPCR results showing the relative enrichments of the indicated m^6A readers among the targeted genes in mouse oocytes. The data validated by qPCR are present as mean \pm SD and are normalized to IgG samples. P value, unpaired t test.

f-i. RIP-qPCR results showing the relative enrichments of IGF2BP2 among the targeted genes in WT and *Alkbh5*^{-/-} MII oocytes. The data validated by qPCR are present as mean \pm SD and are normalized to IgG samples. P value, unpaired t test.

This study performed low-input m^6A meRIP-seq on approximately 2800 GV oocytes from WT and *Alkbh5*^{-/-} female mice. The effort of collecting so many oocytes for study is appreciated, but there is only one replicate, which is not good for the quality of the dataset. Recently, at least three low-input m^6A meRIP-seq datasets in mouse oocytes and early embryos have been published. Can the authors compare their dataset with the published datasets to see whether they are consistent with each other? Can the authors only select those bona fide m^6A sites to compare between WT and *Alkbh5*^{-/-} oocytes, and use these sites for downstream analysis?

Reply:

Thank you for the kind suggestions. To improve our data, we compared our m^6A -seq data from 2800 GV oocytes with recent picoMeRIP-seq data from 49 GV oocytes (2 replicates) (PMID: 37081317) and another merged scm^{6A}-seq dataset from single GV oocytes (10 replicates) (PMID: 36658155) using the same criterion. We found that nearly 85.5% of m^6A -marked transcripts in picoMeRIP-seq, approximately 68.13% in scm^{6A}-seq and 92.99% of methylated transcripts detected in both datasets were also identified in our datasets (**Rebuttal Fig. 1a and Revised Supplementary Fig. 5d**). Furthermore, we conducted GO analysis and found that commonly modified transcripts are greatly

enriched in transcription and cell cycle functions, which is consistent with the existing findings (**Rebuttal Fig. 1b and Revised Supplementary Fig. 5e**). Transcripts of key genes for meiotic resumption and transcription were detected to have m⁶A modifications in GV oocytes and were highly represented among all three datasets, as exemplified by IGV snapshots (**Rebuttal Fig. 1c and Revised Supplementary Fig. 5f**).

Rebuttal Fig 1. Comparative analysis of m⁶A-seq datasets.

a. Venn diagram showing the overlaps of identified m⁶A-modified RNAs in GV oocytes from picoMeRIP-seq, scm⁶A-seq and our datasets.

b. GO analysis of commonly identified m⁶A-marked RNAs by different methods. Pos. positive; neg. negative; reg. regulation.

c. IGV snapshots showing m⁶A distributions of the representative conserved m⁶A-modified genes (*Mos*, *Mastl*, *Cebpb*, *Foxo1*) detected by different methods. The sample-specific scale ranges are indicated in brackets.

We designated a gene as m⁶A⁺ if it was identified in both our dataset and at least one of the other datasets; otherwise, it was designated as m⁶A⁻. Thus, we identified 6263 transcripts that carried m⁶A modification among the 14404 expressed genes (FPKM>1 in at least one sample). As suggested by the reviewer, we then performed downstream analysis on this set of genes. Integrative analysis with an RNA-seq dataset revealed that m⁶A⁺ transcripts decay more slowly than m⁶A⁻ transcripts (**Rebuttal Fig. 10a and Revised Supplementary Fig. 5g**). We also observed that *Alkbh5* depletion depressed the degradation of both m⁶A⁺ and m⁶A⁻ transcripts, with m⁶A⁺ transcripts showing the slowest decay (**Rebuttal Fig. 10b, 10c and Revised Supplementary Fig. 5h, 5i**). Consistently, increased RNA abundance of m⁶A⁺ genes was observed in KO MII oocytes (**Rebuttal Fig. 10d and Revised Supplementary Fig. 5j**). These observations were reminiscent of the findings derived from our m⁶A-seq dataset.

Collectively, these results indicated the reliability of our m⁶A-seq data.

Rebuttal Fig 10. Downstream analysis of integrated m⁶A⁺ transcripts.

a. Cumulative fraction and boxplot (inset) showing the difference in degradation rates (\log_2 [fold change (MII/GV)]) between the integrated m⁶A-modified and unmodified RNAs in WT oocytes. P values, Wilcoxon signed-rank test.

b-c. Cumulative fraction and boxplot (inset) showing the degradation rates (\log_2 [fold change (MII/GV)]) of the integrated m⁶A-modified and unmodified RNAs in WT and *Alkbh5*^{-/-} oocytes, respectively. P values, Wilcoxon signed-rank test.

d. Violin plot showing the expressional levels of the integrated m⁶A-modified transcripts in WT and *Alkbh5*^{-/-} oocytes at MII stage. The data are presented as violin with central line denoting mean value. P values, Mann-Whitney test.

Line 571~572: A total of 2450 m6A peaks were differentially changed upon *Alkbh5* knockout, with 1325 upregulated and 1125 downregulated enrichments. While *Alkbh5* is expected to remove m6A, how does the authors explain the observation of so many (1125 out of 2450) m6A peaks that are downregulated upon *Alkbh5* knockout?

Reply:

Thank you for making this point clear. In Fig. S5 (Fig. 6 in the revised version), we showed that there were 1125 hypomethylated and 1325 hypermethylated m⁶A peaks in KO GV oocytes. Considering the role of *Alkbh5* in demethylation, the proportion was unexpectedly high for hypomethylated peaks and relatively low for hypermethylated peaks, as noted by the reviewer. We believe this phenomenon could be justified by the following two points:

- 1) **The impact of ALKBH5-mediated demethylation on maternal RNAs is limited before GVBD, since ALKBH5 is restricted to the nucleus.** Maternal RNAs (which are localized to the cytoplasm) could not undergo m⁶A removal by ALKBH5 because this process occurs in the nucleus after transcription and before RNA export to the cytoplasm. The nuclear localization of ALKBH5 and the life cycle of m⁶A RNA have been described in many preceding papers. In this review (PMID: 31520073), the author interpreted this by saying that “m⁶A is an imprint of nuclear events...” Thus, the function of ALKBH5 seems to be depressed (which is also indicated by minor transcriptome changes in KO GV oocytes) in GV oocytes, where transcription has ceased and maternal RNAs are already stored in the cytoplasm.
- 2) **Another m⁶A eraser, FTO, may have a compensatory function for ALKBH5 in removing m⁶A deposition in GV oocytes.** In contrast to the nuclear localization of ALKBH5, FTO is distributed in the cytoplasm of GV oocytes, as shown by the immunostaining results of mouse ovarian follicles (PMID: 37165445), suggesting the potential role of FTO in GV oocytes. This notion was also supported by an excellent study showing that *Fto*^{-/-} females exhibited a lower ratio of surrounding nucleus GV oocytes but increased chromosome and spindle abnormalities

at the MII stage (PMID: 35511947).

In conclusion, the high proportion of hypomethylated peaks is presumably due to the limited impact of ALKBH5 and the synergistic function of ALKBH5 and FTO in m⁶A removal in mouse oocytes.

This is a newly generated *Alkbh5*^{+/-} mouse model by CRISPR. What are the phenotypes of the *Alkbh5*^{-/-} mice? Are they the same as previously reported *Alkbh5*^{-/-} mouse model? Is the homozygous rate less than 25% in the *Alkbh5*^{+/-} x *Alkbh5*^{+/-} mating? Are the size of *Alkbh5*^{-/-} male or female mice smaller than their *Alkbh5*^{+/+} and *Alkbh5*^{+/-} littermates? Although this study focused on oocytes from *Alkbh5*^{-/-} female mice, the authors should provide more detailed phenotypes of this new mouse line.

Reply:

Thank you for the helpful suggestions. We apologize for the lack of a detailed description of our new mouse line. Despite the different generation methods, **the phenotypes of *Alkbh5*^{-/-} mice in our present study largely resemble those of *Alkbh5*^{-/-} mice previously reported by Zheng (PMID: 23177736), in which *Alkbh5* was deleted by the LoxP/Cre system.** In our current study, *Alkbh5*^{-/-} mice were viable, anatomically normal, and smaller in size than their *Alkbh5*^{+/+} and *Alkbh5*^{+/-} littermates (**Rebuttal Fig. 11a and Revised Supplementary Fig. 2c**). As reported previously, smaller testes were observed in *Alkbh5*^{-/-} males than in their WT littermates.

When heterozygous *Alkbh5*^{+/-} mice were crossed, **we found that *Alkbh5*^{-/-} mice were retrieved in sub-Mendelian ratios (20% homozygous vs. 25% expected) (Rebuttal Fig. 11b and Revised Supplementary Fig. 2d).**

We have added these data to **Revised Supplementary Fig. 2** and modified the corresponding text (**Page 15, lines 421-425**).

Rebuttal Fig 11. Phenotypes of *Alkbh5*^{-/-} mice.

a. Representative image of WT and *Alkbh5*^{-/-} mice of 1 month.

b. Table of numbers and percentages of mice per genotype retrieved from heterozygous intercrosses.

The authors were able to collect some MII oocytes from *Alkbh5*^{-/-} female mice for analysis. Can the *Alkbh5* KO MII oocytes be fertilized by WT sperm? If yes, what are the phenotypes of the *Alkbh5* maternal KO embryos at the preimplantation stages?

Reply:

Thank you for making this point clear. As requested, we superovulated more than three *Alkbh5*^{-/-}

females at 4 weeks and then mated them with WT males. After successful mating, fertilized oocytes were collected from the oviducts 28 h post hCG injection, and the fertilization rate was assessed by examining pronucleus formation. Notably, approximately half of the ovulated oocytes derived from *Alkbh5*^{-/-} females were competent to be fertilized (**Rebuttal Fig. 7a, 7b and Revised Fig. 11, 1m**). To avoid interference from unfertilized oocytes, we next selected fertilized embryos for *in vitro* culture. Approximately 40% of zygotes reached the 2-cell stage, and none developed beyond the 2-cell stage (**Rebuttal Fig. 7a, 7b and Revised Fig. 11, 1m**), suggesting impaired MZT upon maternal *Alkbh5* knockout.

In conclusion, ovulated oocytes from *Alkbh5*^{-/-} females were competent to be fertilized but ultimately arrested at the 1-cell or 2-cell stage.

Rebuttal Fig 7. Embryos derived from *Alkbh5*^{-/-} oocytes were arrested at the 2-cell stage.

a. Representative pictures of embryos derived from WT and *Alkbh5*^{-/-} oocytes at 28, 40 and 54 h after hCG injection.

b. Developmental rates of embryos that reach the indicated stages at corresponding timepoints. The data are presented as mean ± SD. P value, unpaired t test.

Minor points:

The discussion part is too long, need to be cut and make it concise.

Reply:

Thank you for the helpful suggestion. The discussion section has been revised accordingly.

The writing and English of this manuscript could be further polished.

Reply:

Thank you for the suggestion. The manuscript has been polished by a professional English language editing service.

The specificity of antibody is very important. This study used two anti-ALKBH5 antibodies, one for WB, the other one for RIP (Supplementary table S3). Have the authors tested or validated the specificity of these ALKBH5 antibodies? Which anti-ALKBH5 antibody is used for immunohistochemistry in Figure S1A? Which anti-ALKBH5 antibody is used for immunofluorescence to detect the ALKBH5-GFP protein in Figure S1B? Why the authors did not show immunofluorescence results for ALKBH5 in WT oocytes without overexpression? The authors indeed performed immunohistochemistry in *Alkbh5*^{-/-} ovary, but why they did not perform immunofluorescence in *Alkbh5*^{-/-} oocytes?

Reply:

Thank you for making these points clear. In our present study, we used two commercial anti-ALKBH5 antibodies (Sigma and Proteintech) for different purposes. Prior to these experiments, we validated the specificity of these two ALKBH5 antibodies, as shown by the WB results in mGCs (**Rebuttal Fig. 12a**). The antibody from Sigma (Cat. HPA007196) was used for the IHC shown in **Figure S1A** (we have clarified the antibody used for IHC in the Materials section), and the specificity of this antibody for IHC was also confirmed by immunohistochemical staining of *Alkbh5* knockout ovary sections. Both antibodies were available for RIP, and the figure below clearly shows that similar results were obtained from RIP using the two different antibodies (**Rebuttal Fig. 12b, 12c**). Thus, we showed the data of Proteintech antibody in the main figure (**Revised Fig. 7**), as it yielded better outcomes in our case.

Regarding the immunofluorescence results for ALKBH5, we tested four commercial antibodies against ALKBH5 for this paper (Sigma, Proteintech, Abcam and CST) and **found that none could be used for immunostaining**. The signals of these antibodies seem to be nonspecific, as signals were observed in both WT and KO oocytes. Thus, alternatively, we microinjected oocytes with **GFP-tagged *Alkbh5* mRNA** to visualize ALKBH5 distribution without immunofluorescence staining.

Rebuttal Fig 12. Validation of ALKBH5 antibody specificity.

a. WB results showing the ALKBH5 protein levels in WT and *Alkbh5* KO mouse granulosa cells using two different antibodies.

b. RIP-qPCR analysis showing the fold changes of ALKBH5 binding ability with the indicated

RNAs relative to IgG in mouse oocytes using ALKBH5 antibody (Proteintech).

c. RIP-qPCR analysis showing the fold changes of ALKBH5 binding ability with the indicated RNAs relative to IgG in mouse oocytes using ALKBH5 antibody (Sigma).

Line 160~162: Capped mRNAs encoding fluorescently labelled full-length of *Alkbh5* coding region were in vitro transcribed from reconstructed pRK5-Flag-eGFP (a gift from Dr. Hengyu Fan) using mMACHINE™ SP6 kit. Is this pRK5-*Alkbh5*-Flag-eGFP plasmid published before? If yes, please cite the paper. If not, this will be the first time to report this plasmid. The authors should provide the details about the construction this plasmid, such as the primers and method to amplify the full-length *Alkbh5* coding region and cloning strategy.

Reply:

Thank you for the suggestion. The pRK5-*Alkbh5*-Flag-eGFP plasmid was constructed by our lab and reported for the first time. The *Alkbh5* (NM_172943.4) open reading frame sequence was obtained by PCR-based amplification of mouse cDNA with appropriate primers (**Supplementary Table 1**) using a Q5® High-Fidelity DNA Polymerase amplification system (NEB, M0491). The original vector pRK5-Flag-eGFP (a gift from Dr. Hengyu Fan) contains an SP6 promoter that allows *in vitro* transcription and multiple cloning sites. In our case, XhoI and XbaI were selected as cloning sites, and the obtained *Alkbh5* sequence was fused into the vector by restriction digestion and ligation using restriction endonucleases (NEB, R0146V and R0145V) and T4 DNA ligase (NEB, M0202S). The fidelity of the reconstructed vector was confirmed by DNA sequencing. We added a more detailed description of plasmid construction in the Methods section (**Pages 6-7, lines 173-187**).

Line 278~280: The following poly(A) tailing and PCR amplification were performed according to instructions. PCR products were then subjected to enzyme incision for fragmentation. What are the instructions here? What kit did the authors use? What enzyme did authors use for fragmentation? The authors should describe the full details of low input RNA-seq procedure.

Reply:

We thank the reviewer for the constructive comments. Accordingly, the detailed methods have been provided in the Materials and Methods section (**Page 11, lines 304-311; Supplementary Table 3**). The following poly(A) tailing and PCR amplification were performed according to the instructions (PMID: 24385147). Briefly, after the first-strand synthesis, the cDNA was preamplified by 12 cycles of PCR to obtain the full-length cDNA products through IS PCR primer-mediated semisuppressive PCR. Then, the cDNA was fragmented by incubation with dsDNA Fragmentase (NEB, M0348S) at 37°C for 30 min and size-selected by magnetic beads to obtain 150-300 bp fragments. Libraries were constructed with sheared cDNA using the SMART-Seq v4 Ultra Low Input RNA Kit (Takara, Japan) and then sequenced on an Illumina NovaSeq™ 6000 system (PE150).

REVIEWERS' COMMENTS

Reviewer #1 (Remarks to the Author):

I appreciate the authors' efforts in addressing the questions and points raised by me and other reviewers. I am pleased with the additional work done, which has significantly improved the manuscript's quality and strengthened its scientific significance. I have no further questions.

Reviewer #2 (Remarks to the Author):

The authors have adequately addressed all my concerns, and I recommend this manuscript to be published.

Reviewer #3 (Remarks to the Author):

The Alkbh5 KO phenotype reported here is interesting. The m6A MeRIP-seq performed in oocytes are valuable. These are all good results to show.

The authors did not really identify comprehensively substrates of ALKBH5. This is a limit that the authors should clearly state.

I remain not convinced by the IGF2BP data, they used CLIP-seq from ESCs to infer mouse oocytes. These are completely different systems. At the best they can test one or two target and suggest those go through IGF2BP pathway. They cannot make any broad claims.

Point-by-point response to the reviewers' comments

Reviewer #1 (Remarks to the Author):

I appreciate the authors' efforts in addressing the questions and points raised by me and other reviewers. I am pleased with the additional work done, which has significantly improved the manuscript's quality and strengthened its scientific significance. I have no further questions.

Reply:

We thank Reviewer #1 for positive comments and recognition of our efforts in addressing the questions.

Reviewer #2 (Remarks to the Author):

The authors have adequately addressed all my concerns, and I recommend this manuscript to be published.

Reply:

We thank Reviewer #2 for positive comments and support.

Reviewer #3 (Remarks to the Author):

The Alkbh5 KO phenotype reported here is interesting. The m6A MeRIP-seq performed in oocytes are valuable. These are all good results to show.

Reply:

We thank Reviewer #3 for positive comments and further guidance to improve our manuscript.

The authors did not really identify comprehensively substrates of ALKBH5. This is a limit that the authors should clearly state.

Reply:

Thank you for the suggestion. Accordingly, we have toned down the claim on ALKBH5 substrates in the revised manuscript (**Pages 13-15, lines 376, 395-396, 402-403, 432**) and stated this limitation in the discussion section (**Page 19, lines 563-568**).

I remain not convinced by the IGF2BP data, they used CLIP-seq from ESCs to infer mouse oocytes. These are completely different systems. At the best they can test one or two target and suggest those go through IGF2BP pathway. They cannot make any broad claims.

Reply:

Thank you for the suggestion. We have turned down the broad claims on IGF2BP2 function as suggested by the reviewer, and mainly focused on the targeted transcripts that we have testified in the revised manuscript (**Pages 4, 16 and 18; lines 105-106, 457, 474-475, 515-516**). We also discussed this point in our discussion section (**Page 20, lines 587-590**).